# EvoAgent: Self-evolving Agent with Continual World Model for Long-Horizon Tasks

## Abstract

Completing Long-Horizon (LH) tasks in open-ended worlds is an important yet difficult problem for embodied agents. Existing approaches suffer from two key challenges: (1) they heavily rely on experiences obtained from human-created data or curricula, failing to autonomously update and select multimodal experiences, and (2) they may encounter catastrophic forgetting issues when faced with new tasks, failing to autonomously update world knowledge. To solve these challenges, this paper presents *EvoAgent*, a self-evolving agent with a continual World Model (WM), which can autonomously complete various LH tasks across environments through self-planning, self-control, and self-reflection, without human intervention. Our proposed EvoAgent contains three modules, i.e., i) the memory-driven planner which uses an LLM along with the WM and interaction memory, to convert LH tasks into executable sub-tasks; ii) the WM-guided action controller which leverages WM to generate low-level actions and incorporates a self-verification mechanism to update multimodal experiences; iii) the experience-inspired reflector which implements a two-stage curriculum learning algorithm to select experiences for task-adaptive WM updates. Moreover, we develop a continual World Model for EvoAgent, which can autonomously update the multimodal experience pool and world knowledge through closed-loop dynamics. We conducted extensive experiments on Minecraft and Atari, compared with existing methods, EvoAgent can achieve an average success rate improvement of 111% and reduce ineffective actions by more than 6x.

## 1 Introduction

Long-horizon (LH) tasks (Shen et al., 2025; Guo et al., 2024) are complex, multi-step tasks that require sustained planning, sequential decision-making, and extended execution over a prolonged period to achieve a final goal. These tasks are challenging, often exhibiting reward sparsity (Hafner et al., 2025) and procedural diversity (Yang et al., 2024). Completing LH tasks in open-ended worlds is an important yet difficult problem for embodied agents, such as logistics robots (Luo et al., 2025), surgical robots (Marcus et al., 2024), and rescue robots (Jadeja et al., 2024).

On the one hand, existing agents have made remarkable progress by utilizing expert data and domain-specific curricula created by humans, developing policies through Reinforcement Learning (RL) (Ren et al., 2025; Mazzaglia et al., 2024b), Imitation Learning (IL) (Liu et al., 2024), and Large Language Models (LLMs) (Li et al., 2025). On the other hand, recent studies (Kwa et al., 2025) demonstrate that humans' ability to accomplish LH tasks in an open world relies on autonomous experience accumulation and world knowledge updates. In essence, autonomous world knowledge update serves as a meta-cognitive driver that not only guides action selection under partial observability but also enables context-aware adaptation to environmental dynamics, thereby resolving the local optimality issue inherent in LH task completion.

*Completing long-horizon tasks in open-ended worlds requires embodied agents to achieve autonomous experience accumulation and world knowledge updates, like a baby thrives.*

Nevertheless, existing methods are hard to complete various LH tasks across environments from scratch: *1) Failing to autonomously update and select multimodal experiences.* Most embodied agents assume that all training data are available from the beginning (such as IL-based or LLMs-based agents), which heavily rely on human-created data or curricula (Li et al., 2025). However, this

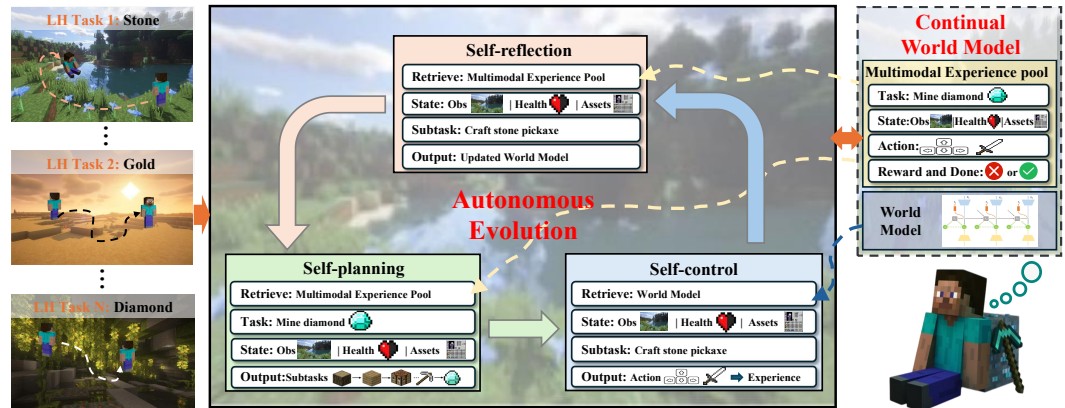

Figure 1: EvoAgent, a self-evolving agent with a continual World Model (WM). Take Minecraft as an example. **Left:** Various Long-Horizon (LH) tasks across environments. **Middle:** EvoAgent can autonomously complete various LH tasks across environments by self-planning, self-control, and self-reflection, without human intervention. **Right:** We build a continual WM for EvoAgent. Through closed-loop dynamics, EvoAgent can autonomously update the multimodal experience pool and world knowledge.

assumption is unrealistic, as agents may encounter novel tasks or environments after deployment (Zhang et al., 2024). *2) Failing to autonomously update world knowledge.* Existing methods use LLMs (such as Voyager (Wang et al., 2023a), Jarvis-1 (Wang et al., 2023c)) to represent world knowledge based on sampling historical experiences or use a graph (such as Optimus-1 (Li et al., 2024b)) to sparsely represent world knowledge, which requires human intervention and is hard to autonomously update. Existing methods face catastrophic forgetting, where they lose previously obtained knowledge (Nayak et al., 2025; Hafner et al., 2025) for learning new tasks, which are hard to autonomously update and transfer world knowledge for LH tasks across environments.

To solve this problem, in this paper, we propose **EvoAgent** (as shown in Figure 1), a self-evolving agent with a continual World Model (WM), which can autonomously complete various LH tasks across environments through self-planning, self-control, and self-reflection, without human intervention. Our proposed EvoAgent contains three modules: i) The experience-driven task planner, which uses an LLM along with interaction experiences, to incorporate self-state into the planning phase and convert LH tasks into executable sub-tasks; ii) The WM-guided action controller, which leverages WM to generate low-level actions and incorporates a self-verification mechanism to update multimodal experiences. iii) The Curriculum Learning (CL) -based reflector, which implements a two-stage CL algorithm to select experiences for task-adaptive WM updates. Moreover, we propose a novel continual WM for EvoAgent as well. By utilizing a model-based online RL setup and closed-loop dynamics, EvoAgent is able to autonomously update the multimodal experience pool and world knowledge, filtering out invalid explorations and mitigating historical forgetting.

We evaluate EvoAgent's performance in Minecraft (Fan et al., 2022), a popular open-world environment. Extensive experiments demonstrate EvoAgent's superiority: compared with existing methods, EvoAgent can achieve an average success rate improvement of 111% and reduce ineffective actions by more than 6x. Ablation studies confirm that our Continual WM contributes 72% of the performance gain by enabling coherent knowledge integration. We also evaluate the generalization of EvoAgent in the Atari environment (Bellemare et al., 2013). The contributions of this paper are summarized as follows:

- We propose EvoAgent, which can autonomously complete various LH tasks across various environments through self-planning, self-control, and self-reflection, without human intervention.

- We build a novel continual WM for EvoAgent, which can autonomously update the multi-modal experience pool and world knowledge through closed-loop dynamics.

- We conduct extensive experiments on Minecraft and Atari to validate the superiority of EvoAgent, where the proposed EvoAgent can achieve an average success rate improvement of 111% and reduce ineffective actions by more than 6x compared with existing methods.

## 2 RELATED WORKS

**Embodied agents solving long-horizon tasks.** Long-Horizon (LH) tasks (Shen et al., 2025; Guo et al., 2024; Chen et al., 2024) refer to complex, multi-step tasks. Existing work on embodied agents completing LH tasks can be divided into two categories. One is Model-Based Reinforcement Learning (MBRL) (Mazzaglia et al., 2024a). Embodied agents leverage MBRL to tackle LH tasks by interacting with environments and learning predictive world dynamics (Liu et al., 2024). Such as GenRL (Mazzaglia et al., 2024b) proposes a multimodal-foundation model that aligns vision-language representations with generative world dynamics for RL. The other is vision-language model-based (VLM) planning (Roger et al., 2025). Embodied agents leverage VLMs to decompose LH tasks into hierarchical sub-goals (Liu et al., 2024), dynamically refine plans via memory-augmented reasoning (Song et al., 2024), and align semantic intent with executable actions through iterative simulation (Yang et al., 2024), such as EmbodiedGPT (Mu et al., 2023), which bridges high-level planning with low-level control. However, they assume perfect knowledge of environments, rely on oracle feedback, and assume perfect execution of low-level policies, which makes it hard to adapt various LH tasks across environments in open worlds (Zhang et al., 2024).

**World Model (WM).** WMs are foundational blocks of AI systems to perform planning and reasoning (Ha & Schmidhuber, 2018). They serve as simulators of real environments that predict the future outcome of certain actions, and policies can be derived from them. Current research focuses on two paradigms: understanding the world through latent state representations (Hansen et al., 2023; Zhou et al., 2024) and predicting future dynamics for planning and control (Ma et al., 2024; Wang et al., 2024). Representative example usages of them in MBRL include action searching (Nayak et al., 2025; Schrittwieser et al., 2020), policy optimization within such simulators (Feinberg et al., 2018; Hafner et al., 2019a), or a combination of both (Hafner et al., 2025; Chitnis et al., 2023). However, WMs currently struggle to prevent catastrophic forgetting (Mattes et al., 2023) due to their inability to maintain stable representations of previously learned environmental dynamics while adapting to new tasks, often exacerbated by shared parameter updates prior to knowledge (Sun et al., 2024).

## 3 EVOAGENT

**Framework.** Let $\mathcal{E}$ denote a dynamic open-world environment with partial observability, $\mathcal{T}$ represent the long-horizon tasks, and $\mathcal{S}$ represents the agent's current state. We aim to design a self-evolving agent *EvoAgent* that can complete various long-horizon tasks across environments, without human intervention. As shown in Figure 2, EvoAgent includes an experience-driven task planner $\Psi_{\text{plan}}$, a WM-guided action controller $\Pi_{\text{act}}$, a CL-based reflector $\Phi_{\text{reflect}}$; The continual world model includes a Multimodal Experience Pool (MEP) $\mathcal{D}_{\text{MEP}}$, and a world model $\mathcal{M}_w$. EvoAgent has a model-based online RL setup and can be instantiated as:

$$EvoAgent : \langle \Psi_{\text{plan}}, \Pi_{\text{act}}, \Phi_{\text{reflect}}, \mathcal{D}_{\text{MEP}}, \mathcal{M}_w \rangle \tag{1}$$

**Continual world model.** As shown in Algorithm 1, the task planner, the action controller, and the reflector are three separate modules, linked together by a continual world model. EvoAgent with a continual world model can self-evolve through closed-loop dynamic self-planning, self-control, and self-reflection, autonomously updating the multimodal experience pool and world knowledge, The sketch of EvoAgent is as follows:

$$\overbrace{\underbrace{\textbf{Planner}}_{\substack{\Psi_{\text{plan}} \triangleright \mathcal{D}_{\text{MEP}} \\ \downarrow \{g_i\}}} \rightarrow \underbrace{\textbf{Controller}}_{\substack{\Pi_{\text{act}} \circ \mathcal{M}_w \\ \downarrow \{a_t\}, \mathcal{D}_{\text{MEP}}}} \rightarrow \underbrace{\textbf{Reflector}}_{\substack{\Phi_{\text{reflect}} \triangleright \mathcal{D}_{\text{MEP}} \\ \downarrow \theta'_{\mathcal{M}_w}}} \rightarrow}^{\mathcal{E}, \mathcal{T}, \mathcal{S}, \mathcal{D}_{\text{MEP}}, \mathcal{M}_w} \tag{2}$$

where $\{g_i\}$ are subtasks generated by the planner $\Psi_{\text{plan}}$; $\{a_t\}$ are actions generated by the controller $\Pi_{\text{act}}$; $\theta'_{\mathcal{M}_w}$ is the updated parameter of the world model $\mathcal{M}_w$.

**Evaluation.** According to relevant research (Hafner et al., 2025; Guo et al., 2024), the agents' performance evaluation includes Success Rate (**SR**) and Exploration Efficiency (**EE**).

$$SR = \frac{\text{Episode}_{g_i}^{\text{suc}}}{\text{Episode}^{\text{all}}}, EE = \frac{\mathcal{L}_{g_i}^{\text{suc}}}{\mathcal{L}_{g_i}^{\text{all}}} \tag{3}$$

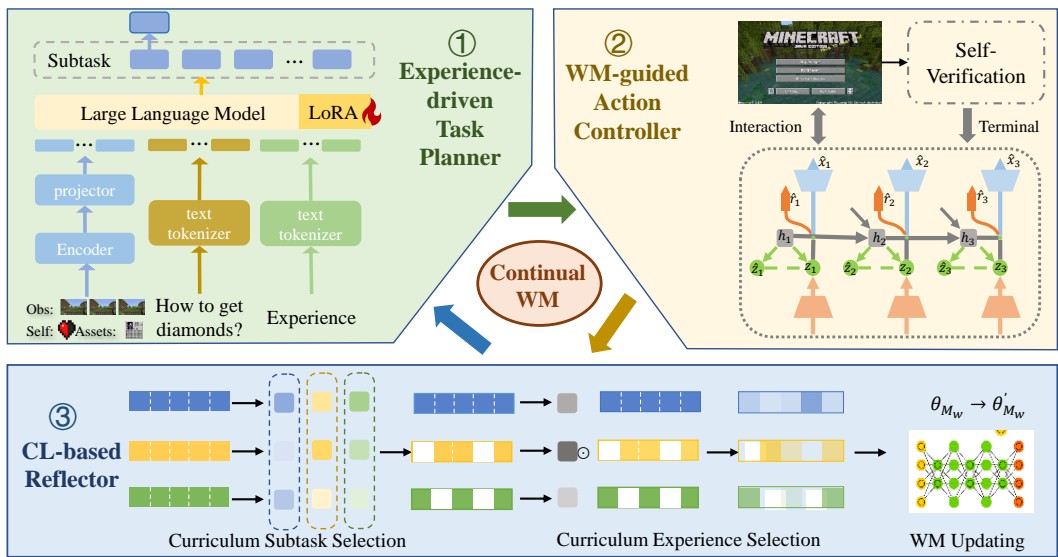

Figure 2: EvoAgent Framework, which includes three modules empowered by a continual WM.

where $\text{Episode}_{g_i}^{\text{suc}}$ indicates the number of episodes in which the subtask $g_i$ succeeded; $\text{Episode}_{g_i}^{\text{all}}$ indicates the total number of episodes; $L_{g_i}^{\text{suc}}$ indicates the success step length of subtask $g_i$, and $L_{g_i}^{\text{all}}$ indicates the total step length of subtask $g_i$ exploration.

## 3.1 PRELIMINARIES

**World model.** Recurrent State-Space Model (RSSM) (Hafner et al., 2025; 2019b) is a classic world model structure, which can predict latent states and rewards from high-dimensional observations. RSSM contains 6 modules. 1) Encoder, maps observation $o_t$ to a stochastic latent state $s_t = (h_t, z_t)$, where $h_t$ is a deterministic RNN state and $z_t$ is a stochastic latent variable, $q_\phi(z_t|h_t, o_t) = \mathcal{N}(z_t; \mu_\phi(h_t, o_t), \sigma_\phi(h_t, o_t))$, where $\mu_\phi, \sigma_\phi$ are neural networks. 2) Sequence model: predicts the sequence of these representations given past actions $a_{t-1}$, $h_t = f_\theta(h_{t-1}, z_{t-1}, a_{t-1})$. 3) Dynamics predictor, predicts the prior latent state transition, $p_\theta(\hat{z}_t|h_t) = \mathcal{N}(\hat{z}_t; \mu_\theta(h_t), \sigma_\theta(h_t))$. 4) Decoder: reconstructs observations from latent states, $p_\theta(o_t|h_t, z_t) = \mathcal{N}(o_t; \mu_\theta^{\text{obs}}(h_t, z_t), \sigma_\theta^{\text{obs}})$. 5) Reward predictor, predicts rewards, $\hat{r}_t = r_\theta(h_t, z_t)$. 6) Continual predictor, predicts episode continuation flags, $\hat{c}_t = \text{sigmoid}(c_\theta(h_t, z_t))$. Above all, RSSM can be defined as follows:

$$\text{Encoder:} \qquad z_t \sim q_\phi(z_t|h_t, o_t) \qquad (4)$$

$$\text{Sequence model:} \qquad h_t = f_\theta(h_{t-1}, z_{t-1}, a_{t-1}) \qquad (5)$$

$$\text{Dynamics predictor:} \qquad \hat{z}_t \sim p_\theta(\hat{z}_t|h_t) \qquad (6)$$

$$\text{Decoder:} \qquad \hat{o}_t \sim p_\theta(\hat{o}_t|h_t, z_t) \qquad (7)$$

$$\text{Reward predictor:} \qquad \hat{r}_t \sim r_\theta(\hat{r}_t|h_t, z_t) \qquad (8)$$

$$\text{Continual predictor:} \qquad \hat{c}_t \sim c_\theta(\hat{c}_t|h_t, z_t) \qquad (9)$$

**Online model-based reinforcement learning.** As shown in Appendix A.

## 3.2 EXPERIENCE-DRIVEN TASK PLANNER

The experience-driven task planner $\Psi_{\text{plan}}$ is formalized as a function that maps the current multi-modal state $\mathcal{S}$, long-horizon task $\mathcal{T}$, and experience $\mathcal{D}_{\text{MEP}}$ to a sequence of subtasks $\mathcal{G}$.

$$\Psi_{\text{plan}} : \mathcal{S} \times \mathcal{T} \times \mathcal{D}_{\text{MEP}} \to \mathcal{G} \qquad (10)$$

$$\mathcal{S} = \mathcal{O}_{\text{obs}} \times \mathcal{S}_{\text{self}} \times \mathcal{S}_{\text{assets}}, s_t \in \mathcal{S} \qquad (11)$$

$$\mathcal{D}_{\text{MEP}} = \{h\}, h = \langle (s_t, a_t, r_t, s_{t+1}), \mathbb{P}_{(g_i)}|g_i \rangle \qquad (12)$$

where $\mathcal{G} = \{g_i\}_{i=1}^n$ is the subtask space, each subtask $g_i$ satisfies $\bigcup_{i=1}^n g_i \supseteq \mathcal{T}$; $\mathcal{O}_{\text{obs}}$ represents first-person observations, $\mathcal{S}_{\text{self}}$ represents the agent's internal state, such as health or hunger, and $\mathcal{S}_{\text{assets}}$ represents agent's asset library, such as tools; $s_t$ represents multimodal state at step $t$; $h$ represents the experience; $r_t$ represents the reward obtained by performing action $a_t$ at state $s_t$; $\mathbb{P}(g_i)$ indicates the percentage of subtask $g_i$ completion.

As shown in Figure 2, we adopt the image tokenizer $f_v$ to encode the raw images $\mathcal{O}_{obs}, \mathcal{S}_{\text{self}}, \mathcal{S}_{\text{assets}}$ into token embeddings $\mathcal{V} = \{v_1, v_2, ..., v_n\} \in \mathbb{R}^{n \times d}$, where $n$ denotes the number of visual tokens and $d$ is the dimensionality of each token. We adopt the textual tokenizer $f_t$ to encode $\mathcal{T}$ into token embeddings. We further utilize a lightweight projection module $f_l$ with a trainable projection matrix $W$. This module maps the visual tokens to the same space with text embeddings $\hat{\mathcal{V}} = W\mathcal{V}$, yielding $\hat{\mathcal{V}} = \{\hat{v}_1, \hat{v}_2, ..., \hat{v}_n\} \in \mathbb{R}^{n \times d}$. The output of our planner is the subtask $g_i$.

Agents can update the LLM for efficient long-horizon task planning without human intervention as the agent interacts dynamically with the environment. The LLM-based planner undergoes lightweight fine-tuning using Low-Rank Adaptation (LoRA) to enhance its adaptability to encountered challenges. LoRA adapters are injected into the attention modules of the LLM, and only these low-rank parameters are updated during training, keeping the original LLM parameters frozen. 1) During agent initialization, the fine-tuning process utilizes all accumulated experiences from the multimodal experience pool $\mathcal{D}_{\text{MEP}}$ for task planning. When the multimodal experience pool is empty, the agent initializes task planning based on the capabilities of the original GPT-4o. 2) During the agent's lifecycle, when the WM-guided action controller feedback indicates the subtask $g_i$ failure, experience trajectories relevant to the subtask $g_i$ by label matching are extracted to construct input-output pairs $\{(X_{\text{in}}^{(k)}, Y_{\text{out}}^{(k)})\}$ for model fine-tuning, where the input $X_{\text{in}}^{(k)}$ includes all the experience $h$ related the subtask $g_i$, while the output $Y_{\text{out}}^{(k)}$ represents the corresponding subtask sequence. The optimization objective is to maximize the cross-entropy loss between the predicted and history subtask sequences. This enables the planner to quickly study from the failure patterns while preserving its general planning capabilities, thereby improving robustness and reducing repeated errors in long-horizon tasks. 3) When the agent dies (health value is 0), the agent is reinitialized.

## 3.3 WM-GUIDED ACTION CONTROLLER

The WM-guided action controller $\Pi_{\text{act}}$ is formalized as a function that maps the current multimodal state $\mathcal{S}$, subtask $\mathcal{G}$, and the world model $\mathcal{M}_w$ to an action sequence $a_{t:t+H} = \{a_t, a_{t+1}, \ldots, a_{t+H}\}$ for horizon $H$.

$$\Pi_{\text{act}} : \mathcal{S} \times \mathcal{G} \times \mathcal{M}_w \to \mathcal{A} \tag{13}$$

**Action selection.** The controller utilizes $\mathcal{M}_w$ to predict future states and optimize actions:

$$a_{t:t+H} = \underset{a_{t:t+H} \in \mathcal{A}^H}{\arg\max} \mathbb{E}_{\mathcal{M}_w} \left[ \sum_{\tau=t}^{t+H} \gamma^{\tau-t} R(s_\tau, a_\tau, g_i) \right] \tag{14}$$

where $R(s_\tau, a_\tau, g_i)$ is the goal-aligned reward function, and $\gamma \in [0, 1]$ is the discount factor. $R(s_\tau, a_\tau, g_i)$ considers not only the deterministic latent state $h_t$ and stochastic latent variable $z_t$ based on the current observation states $s_\tau$ and actions $a_\tau$ but also a goal embedding $\text{Emb}g_i$ derived from the current subtask $g_i$, which is an extension of the DreamerV3 reward. At each time step $t$, we sample a population of $N$ action sequences $\{a_{t:t+H}^{(k)}\}_{k=1}^N$ from the action space $\mathcal{A}^H$. The world model $\mathcal{M}_w$ is used to predict the future states and compute the expected cumulative reward $\mathbb{E}_{\mathcal{M}_w} \left[ \sum_{\tau=t}^{t+H} \gamma^{\tau-t} R(s_\tau, a_\tau, g_i) \right]$ for each sequence. The sequence $a_{t:t+H}^*$ with the highest rewards is selected, and the first action $a_t^*$ of this sequence is executed in the environment.

**Self-verification.** After executing $a_t$, the agent interacts with $\mathcal{E}$ to collect environment feedback. Then it uses a self-verification mechanism to determine whether the subtask $g_i$ can be terminated. By using this mechanism, agents can reduce inefficient exploration and achieve efficient task execution without human intervention. Let $\phi_{\text{verify}} : \mathcal{S} \times \mathcal{G} \times \mathbb{T} \to \{\text{Terminal}, \text{N-Terminal}\}$ denote the self-verification module, where:

$$\phi_{\text{verify}}(s_t, g_i, t) = \begin{cases} \text{Terminal} & \text{if } \cos(\text{Emb}_{s_t}, \text{Emb}_{g_i}) \geq \sigma \vee t \geq T_{\max} \\ \text{N-terminal} & \text{otherwise} \end{cases} \tag{15}$$

where $\text{Emb}_{s_t}$ is the WM-encoded latent representation of the current state, $\text{Emb}_{g_i}$ is the task embedding derived from the subtask description. Similar to MINEDOJO (Fan et al., 2022), we trained a contrastive video-language model pre-trained on the multimodal experience pool. It computes the cosine similarity $\cos(\cdot)$ between an open-vocabulary language goal embedding $\text{Emb}_{g_i}$ and an 8-frame video snippet embedding $\text{Emb}_{s_t}$, which is used to measure goal attainment with threshold $\sigma$ set empirically. $T_{\max}$ is the maximum allowed steps of each episode. When a subtask $g_i$ is completed or the subtask $g_i$ completion cycle exceeds the maximum step length $T_{\max}$, the subtask $g_i$ is terminated and the experience-driven task planner is performed again.

**MEP Updating.** If the subtask $g_i$ is terminated, whether it is successful or exceeds the step threshold, $\left\{ \langle s_t, a_t, r_t, s_{t+1}, \mathbb{P}_{(g_i)}|g_i \rangle \right\}_{t=0}^{\tau}$ is added to the multimodal experience pool $\mathcal{D}_{\text{MEP}}$.

$$\mathcal{D}_{\text{MEP}} \leftarrow \mathcal{D}_{\text{MEP}} \cup \left\{ \langle s_t, a_t, r_t, s_{t+1}, \mathbb{P}_{(g_i)}|g_i \rangle \right\}_{t=0}^{\tau} \tag{16}$$

New experiences will be autonomously added to the multimodal experience pool.

### 3.4 CL-BASED REFLECTOR

The CL-based reflector $\Phi_{\text{reflect}}$ is formalized as a function that maps the current multimodal state $\mathcal{S}$, subtask $\mathcal{G}$, and the multimodal experience $\mathcal{D}_{\text{MEP}}$ to update the world model from $\mathcal{M}_w$ to $\mathcal{M}'_w$.

$$\Phi_{\text{reflect}} : \mathcal{S} \times \mathcal{G} \times \mathcal{D}_{\text{MEP}} \times \mathcal{M}_w \rightarrow \mathcal{M}'_w \tag{17}$$

$\Phi_{\text{reflect}}$ employs a two-stage CL algorithm to optimize experience selection, which can enable agents to efficiently update the world model without human intervention as the agent interacts dynamically with the environment.

### A. TWO-STAGE CL ALGORITHM

**Stage 1: curriculum subtask selection.** For candidate subtasks $g_i \in \mathcal{G}$, we use four indicators for curriculum subtask selection: (1) the relevance of the subtask $g_i$ to the current target task $\mathcal{T}_{goal}$; (2) the exploration efficiency of the subtask $g_i$ (ratio of successful step length $L_{g_i}^{\text{suc}}$ to total step length $L_{g_i}^{\text{all}}$); (3) the importance of the subtask $g_i$ (comparing its impact on the current world model $\mathcal{M}_{w,g_i}^{\text{new}}$ and past world model $\mathcal{M}_{w,g_i}^{\text{old}}$); (4) the completion ratio of the subtask $\mathbb{P}_{(g_i)}$.

Therefore, curriculum subtask $g_i$ priority score $\rho(g_i)$ for experience $h = \langle (s_t, a_t, r_t, s_{t+1}), \mathbb{P}_{(g_i)}|g_i \rangle$ can be defined as follows:

$$\tau(g_i) = \underbrace{\lambda_1 \cdot \cos(\text{Emb}_{g_i}, \text{Emb}_{\mathcal{T}_{goal}})}_{\text{Relevance}} + \underbrace{\lambda_2 \cdot \frac{L_{g_i}^{\text{suc}}}{L_{g_i}^{\text{all}}}}_{\text{Efficiency}}$$
$$+ \underbrace{\lambda_3 \cdot \text{KL}\left(\mathcal{M}_{w,g_i}^{\text{old}} \| \mathcal{M}_{w,g_i}^{\text{new}}\right)}_{\text{Importance}} + \underbrace{\lambda_4 \cdot \mathbb{P}_{(g_i)}}_{\text{Completion ratio}} \tag{18}$$

where $\cos(\text{Emb}_{g_i}, \text{Emb}_{\mathcal{T}_{goal}})$ represents the cosine similarity of task embedding. $\lambda_1 + \lambda_2 + \lambda_3 + \lambda_4 = 1$ are balancing coefficients. Finally, in round $k$, $|\mathcal{D}_k^{\text{subtask}}|$ subtasks are selected.

$$\mathcal{D}_k^{\text{subtask}} = \{g_i | \tau(h_i) \geq \rho_k\}, \quad \rho_k = \rho_0 \cdot e^{-c_s k} \tag{19}$$

with $c_s$ controlling curriculum subtask progression rate.

**Stage 2: curriculum experience selection.** For candidate experience $h \in \mathcal{D}_{MEP}$ in selected subtasks $\mathcal{D}_k^{\text{subtask}}$, we use three indicators for curriculum experience selection: (1) the Temporal Difference Error (TD-Error) $\delta_{\text{TD}}(h_j)$, prioritizes experience with high TD-Error, indicating prediction mismatch between current and target world models; (2) the Gradient Norm $\|\nabla_{\mathcal{M}_w} \mathcal{L}_{\text{pred}}(h_j)\|$, favors experiences that maximally influence the world model's parameter updates; (3) the Information Gain, measures how much the experience $h_j$ changes the world model's belief distribution, calculated via KL divergence between current $\mathcal{M}_w^{\text{new}}(s_{j+1}|h_j)$ and previous $\mathcal{M}_w^{\text{old}}(s_{j+1}|h_j)$ world model

---

**Algorithm 1** Continual World Model via Closed-Loop Planning-Control-Reflection

---

**Require:** Environment $\mathcal{E}$, Task $\mathcal{T}$, initial MEP $\mathcal{D}_{\text{MEP}}^0$ and $\mathcal{M}_w^0$, Horizon $H$, Max steps $T_{\max}$
**Ensure:** Optimized $\mathcal{D}_{\text{MEP}}^*$, $\mathcal{M}_w^*$
1: Current state $\mathcal{S} \leftarrow (\mathcal{O}_{\text{obs}}, \mathcal{S}_{\text{self}}, \mathcal{S}_{\text{assets}})$
2: **for** Task $\mathcal{T} = \mathcal{T}_0$ to $\mathcal{T}_n$ **do**
3:  $\{g_i\} \leftarrow \Psi_{\text{plan}}(\mathcal{S}, \mathcal{T}, \mathcal{D}_{\text{MEP}})$ {Experience-driven task planner via Eq.(10-12)}
4:   **for** each subtask $g_i \in \{g_i\}$ **do**
5:    **for** episode $t = 1$ to $T_{\max}$ **do**
6:     $\{a_{t:t+H}\} \leftarrow \Pi_{\text{act}}(s_t, g_i, \mathcal{M}_w)$ {WM-guided action controller via Eq.(13-14)}
7:     **if** $\phi_{\text{verify}}(s_t, g_i, t) = $ Terminal **then**
8:      $\mathcal{D}_{\text{MEP}} \leftarrow \mathcal{D}_{\text{MEP}} \cup \{(s_t, a_t, r_t, s_{t+1}, \mathbb{P}_{(g_i)}|g_i)\}$ {MEP updating via Eq.(15-16)}
9:      BREAK
10:     **end if**
11:    **end for**
12:    $\mathcal{D}_k^{\text{subtask}} \leftarrow \text{Curriculum\_Subtask\_Select}(\mathcal{G}_t, \mathcal{T}, \mathcal{D}_{\text{MEP}})$ {CL-based reflector via Eq.(17)}
13:    $\mathcal{D}_k^{\text{exp}} \leftarrow \text{Curriculum\_Experience\_Select}(\mathcal{D}_k^{\text{subtask}})$ {Two-stage CL via Eq.(18-21)}
14:    $\mathcal{M}_w^{'} \leftarrow \Phi_{\text{reflect}}(\mathcal{D}_k^{\text{exp}}, \mathcal{M}_w)$ {WM updating via Eq.(22-23)}
15:   **end for**
16: **end for**

---

predictions.

$$\epsilon(h_j) = \underbrace{\eta_1 \cdot |\delta_{\text{TD}}(h_j)|}_{\text{TD-Error}} + \underbrace{\eta_2 \cdot \|\nabla_{\mathcal{M}_w} \mathcal{L}_{\text{pred}}(h_j)\|_2}_{\text{Gradient Norm}}$$
$$+ \underbrace{\eta_3 \cdot \text{KL}\left(\mathcal{M}_w^{\text{new}}(s_{j+1}|h_j) \| \mathcal{M}_w^{\text{old}}(s_{j+1}|h_j)\right)}_{\text{Information Gain}} \quad (20)$$

where $\eta_1 + \eta_2 + \eta_3 = 1$ are balancing coefficients. Finally, in round $k$, $|\mathcal{D}_k^{\text{exp}}|$ experiences are selected.

$$\mathcal{D}_k^{\text{exp}} = \{h_j | \epsilon(h_j) \geq \rho_k\}, \quad \rho_k = \rho_0 \cdot e^{-c_h k} \quad (21)$$

with $c_h$ controlling curriculum experience progression rate.

## B. WORLD MODEL UPDATING

Update the world model $\mathcal{M}_w$ using experiences $\mathcal{D}_k^{\text{exp}}$ with importance-aware weight $w_j$:

$$\theta'_{\mathcal{M}_w} \leftarrow \theta_{\mathcal{M}_w} - \nabla\left[\underbrace{\sum_{h_j} w_j \mathcal{L}_{\text{pred}}(h_j)}_{\text{Curriculum Loss}} + \underbrace{\mu \cdot \Omega(\theta, \theta^{\text{old}})}_{\text{Regularization}}\right] \quad (22)$$

$$w_j = \frac{\epsilon(h_j)}{\max_k \epsilon(h_k)}, \Omega = \sum_i \mathcal{F}_i(\theta_i - \theta_i^{\text{old}})^2 \quad (23)$$

where $w_j$ to emphasize critical experiences, and $\Omega$ to penalize shifts in parameters critical for past tasks. $\mathcal{F}_i$ is the Fisher information matrix diagonal.

## 4 EXPERIMENTS

### 4.1 EXPERIMENTAL SETTING

**Simulators**. We use Minecraft (Fan et al., 2022) to evaluate EvoAgent. Minecraft features a procedurally generated 3D world of different biomes, which consists of 1-meter-sized blocks that the player and break and place. There are about 30 different creatures that the player can interact with or fight. We employ MineRL 0.4.4 with Minecraft as our simulation environment. The agent operates at a fixed speed of 20 frames per second and only interacts with the environment via low-level control signals. Optimus-1 (Li et al., 2024b) constructs a benchmark of 67 tasks to evaluate the Agent's ability for long-horizon tasks. We use the same task group partitioning as the Optimus-1 to evaluate EvoAgent. We also test the cross-environment generalization of our method in the Atari

Table 1: Main result of EvoAgent. We report the average success rate (SR) and average exploration efficiency (EE) on each task group (as shown in Eq. 3). Upper EE metrics mean that the agent is more efficient at completing the task with fewer invalid exploration steps, while 0.00 indicates that the agent is unable to complete the task. The Overall represents the average result on the three groups of Iron, Gold, and Diamond. The Improving represents the average performance improvement of EvoAgent compared to the algorithms Jarvis-1, dreamerV3, LS-Imagine, and Optimus-1.

| Group | Metric | PPO | GPT-4V | Jarvis-1 | dreamerV3 | LS-Imagine | Optimus-1 | EvoAgent | Improving (%) |
|---|---|---|---|---|---|---|---|---|---|
| Wood | SR↑ | 28.16 | 35.24 | 89.73 | 91.07 | 95.87 | 96.39 | **97.47** | 4.51 |
| | EE↑ | 53.82 | 69.45 | 87.36 | 93.22 | 97.41 | 97.82 | **98.43** | 4.77 |
| Stone | SR↑ | 13.42 | 14.39 | 81.91 | 86.82 | 91.50 | 88.79 | **94.53** | 8.34 |
| | EE↑ | 27.56 | 30.64 | 84.72 | 88.39 | 92.36 | 89.25 | **96.48** | 8.80 |
| Iron | SR↑ | 0.00 | 0.00 | 42.38 | 33.79 | 35.82 | 45.48 | **51.82** | 3.16 |
| | EE↑ | 0.00 | 0.00 | 47.52 | 35.68 | 38.27 | 46.16 | **58.54** | 39.67 |
| Gold | SR↑ | 0.00 | 0.00 | 8.84 | 6.57 | 6.61 | 10.62 | **21.69** | **165.81** |
| | EE↑ | 0.00 | 0.00 | 9.76 | 8.05 | 10.69 | 8.03 | **30.48** | 233.75 |
| Diamond | SR↑ | 0.00 | 0.00 | 7.69 | 4.73 | 4.36 | 9.30 | **17.36** | 166.26 |
| | EE↑ | 0.00 | 0.00 | 0.07 | 3.69 | 4.19 | 7.31 | **26.83** | **603.28** |
| Overall | SR↑ | 0.00 | 0.00 | 19.64 | 15.03 | 15.60 | 21.80 | **30.29** | **111.74** |

simulator. *Atari* (Bellemare et al., 2013) is a cutting-edge, high-fidelity simulation environment for multi-physics analysis and hardware-in-the-loop testing in aerospace and robotics.

**Hyperparameters**. EvoAgent is designed based on the codebase of dreamerV3 (Hafner et al., 2025). The planner of EvoAgent uses the VQ-GAN (Esser et al., 2021) and GPT-4o for task planning. The controller of EvoAgent uses the RSSM-based WM (Hafner et al., 2025) for action selection. EvoAgent runs on a single A100 GPU. Taking $10^7$ steps as an example, compared to dreamerV3 running for 7 days, EvoAgent only needs to run for 2.7 days. For detailed hyperparameters, please refer to the Appendix C. Among them, about the self-verification threshold $\sigma$, we perform a sensitivity analysis by running our agent in Minecraft with $10^7$ environment steps for the task "Iron". As shown in Table 2, experimental results show that the task success rate remains stable when $\sigma \in [0.875, 0.925]$, with sharp declines outside this range due to over/under-termination. When $\sigma < 0.875$, sub-tasks may not be completed but are misjudged, causing subsequent tasks to fail. When $\sigma > 0.925$, due to strict self-verification, sub-tasks may be completed but still require re-planning, reducing the task completion rate.

Table 2: The sensitivity analysis about the hyperparameter $\sigma$.

| $\sigma$ | 0.8 | 0.825 | 0.85 | 0.875 | 0.9 | 0.925 | 0.95 | 0.975 |
|---|---|---|---|---|---|---|---|---|
| Success Rate (SR) | 5.32 | 17.03 | 34.48 | 48.69 | 52.43 | 47.72 | 26.74 | 3.64 |

**LLM API Call**. LLM API calls occur in two processes: subtask planning and subtask failure fine-tuning. The maximum execution steps of each subtask is $T_{max} = 24000$ (Hafner et al., 2025). As the agent self-evolves, the number of subtask failures decreases, which greatly reduces the overhead of subtask failure fine-tuning. Throughout the experiment, with an average of 750 planning calls over $10^7$ environment steps, we spent about $90 to access the GPT-4o API.

**Baselines.** We compare EvoAgent with existing outperforming agents, including model-free Agent (PPO (Schulman et al., 2017)), WM-based agents (dreamerV3 (Hafner et al., 2025), LS-Imagine (Li et al., 2024a)) and LLM-based agents (GPT-4V, Jarvis-1 (Wang et al., 2023d), Optimus-1 (Li et al., 2024b)) on the challenging long-horizon tasks cross-environments. Note that we initialize all agents with an empty multimodal experience pool, while PPO and Jarvis-1 have tools in their initial state. We do not consider agents that are completely based on human data and curricula support (such as Voyager (Wang et al., 2023a), DEPS (Wang et al., 2023b), Steve-Eye (Zheng et al., 2023), and Plan4MC (BAAI, 2023)).

Table 3: Ablation study results. We report the average success rate (SR) on each task group. P.⁻, P., C., R.¹, R.², R., and CWM represent Planning without LoRA, Planning with LoRA, Control, Reflection only with stage 1, Reflection only with stage 2, Reflection with both stages, and Continual World Model, respectively. The PPO algorithm is used by default for model decision-making.

| Setting | | | | | | | Tasks | | | | |
| P.⁻ | P. | C. | R.¹ | R.² | R. | CWM | Wood | Stone | Iron | Gold | Diamond |
|---|---|---|---|---|---|---|---|---|---|---|---|
| ✓ | | | | | | | $28.16_{\pm6.01}$ | $13.42_{\pm7.62}$ | $0.00_{\pm0.00}$ | $0.00_{\pm0.00}$ | $0.00_{\pm0.00}$ |
| | ✓ | | | | | | $41.36_{\pm5.20}$ | $16.27_{\pm8.32}$ | $0.00_{\pm0.00}$ | $0.00_{\pm0.00}$ | $0.00_{\pm0.00}$ |
| | | ✓ | | | | | $45.69_{\pm5.12}$ | $18.37_{\pm6.71}$ | $0.00_{\pm0.00}$ | $0.00_{\pm0.00}$ | $0.00_{\pm0.00}$ |
| | ✓ | ✓ | | | | | $92.42_{\pm3.31}$ | $85.31_{\pm5.96}$ | $31.59_{\pm6.72}$ | $5.47_{\pm2.46}$ | $3.52_{\pm2.31}$ |
| | ✓ | ✓ | ✓ | | | | $93.18_{\pm2.86}$ | $87.72_{\pm5.74}$ | $34.63_{\pm5.19}$ | $8.68_{\pm2.72}$ | $5.14_{\pm2.51}$ |
| | ✓ | ✓ | | ✓ | | | $95.37_{\pm2.48}$ | $91.26_{\pm3.86}$ | $39.58_{\pm5.08}$ | $14.20_{\pm4.05}$ | $8.93_{\pm3.73}$ |
| | ✓ | ✓ | | | ✓ | | $96.69_{\pm2.24}$ | $93.82_{\pm3.34}$ | $42.61_{\pm4.80}$ | $17.53_{\pm5.18}$ | $10.09_{\pm3.54}$ |
| | ✓ | ✓ | | | ✓ | ✓ | $\mathbf{97.47}_{\pm1.75}$ | $\mathbf{94.53}_{\pm2.82}$ | $\mathbf{51.82}_{\pm4.60}$ | $\mathbf{21.69}_{\pm4.61}$ | $\mathbf{17.36}_{\pm2.34}$ |

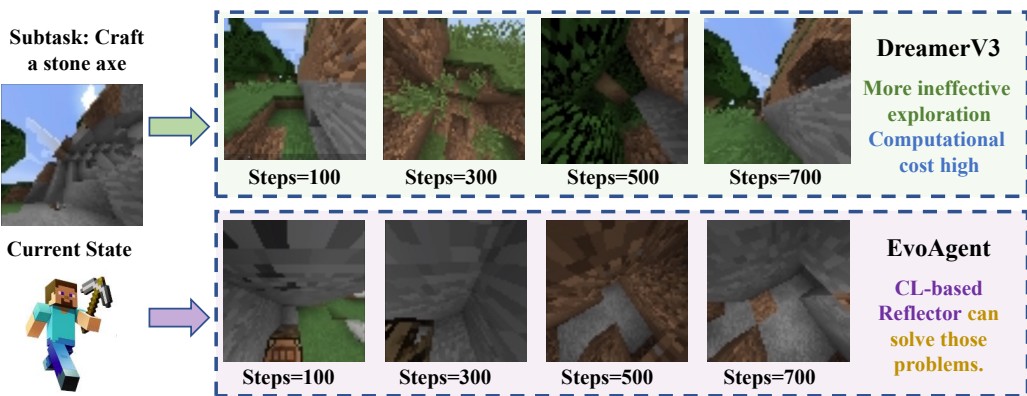

Figure 3: Illustration of the role of CL-based reflector.

## 4.2 EXPERIMENTAL RESULTS

As shown in Table 1, EvoAgent achieves state-of-the-art success rates (SR) and exploration efficiency (EE) across all resource tiers. Compared with existing methods, EvoAgent can achieve an average success rate improvement of 111.74% and reduce ineffective actions by more than 6x (603.28%). For basic tasks (Wood/Stone), EvoAgent marginally outperforms Optimus-1 (97.47% vs. 96.39% SR on Wood) but exhibits significantly greater advantages in advanced tasks like Gold (21.69% vs. 10.62% SR) and Diamond (17.36% vs. 9.30% SR). This hierarchy-aligned improvement suggests EvoAgent's closed-loop planning-control-reflection mechanism effectively addresses long-horizon dependencies, where traditional model-based methods (DreamerV3 and LS-Imagine) and LLM-driven agents (Jarvis-1) struggle to maintain coherent multi-stage strategies. Notably, the EE metric reveals EvoAgent's exploration superiority: its 30.48% EE on Gold tasks is 3.8× higher than Optimus-1, indicating drastically reduced invalid actions during deep resource acquisition.

Model-free methods (PPO) and pure vision-language models (GPT-4V) fail completely (0% SR/EE) on tasks requiring tool hierarchies (Iron+), highlighting their inability to model latent state transitions. While Jarvis-1 and DreamerV3 achieve partial success on intermediate tasks (42.38% SR on Iron), their performance collapses on Gold/Diamond tiers due to compounding errors in action sequences. The 26.83% EE for EvoAgent on Diamond tasks, 7.3× higher than Optimus-1, underscores how CL-based experience selection mitigates exploration bottlenecks in sparse-reward scenarios. LS-Imagine, through a hybrid approach of short-term and long-term imagination, significantly outperforms DreamerV3 in SR and EE metrics on the basic Wood/Stone task. However, its performance growth is slow on complex tasks because accumulated errors can mislead the optimization direction. EvoAgent addresses this issue through a self-evolving Planner-Controller-Reflector approach.

Table 4: For the four metrics in Eq. 18: Relevance (R.), Efficiency (E.), Importance (I.), and Completion Rate (C.R.), we report the average success rate (SR) on the task "Iron" group.

| R. | E. | I. | C.R. | Iron |
|---|---|---|---|---|
| ✓ | | | | 40.16 |
| ✓ | ✓ | | | 41.77 |
| ✓ | ✓ | ✓ | | 46.54 |
| ✓ | ✓ | ✓ | ✓ | 49.37 |

Table 5: For the three metrics in Eq. 20: TD-Error (TD-R.), Gradient Norm (G.N.), and Information Gain (I.G.), we report the average success rate (SR) on the task "Iron" group.

| TD-R. | G.N. | I.G. | Iron |
|---|---|---|---|
| ✓ | | | 42.72 |
| ✓ | ✓ | | 45.59 |
| ✓ | ✓ | ✓ | 50.43 |

Table 6: For the two metrics in Eq. 22: Curriculum Loss (C.L.) and Regularization (R.), we report the average success rate (SR) on the task "Iron" group.

| C.L. | R. | Iron |
|---|---|---|
| ✓ | | 48.61 |
| ✓ | ✓ | 50.92 |

We also test the cross-environment generalization of our method in the Atari simulator. The experimental results are shown in Appendix D.

### 4.3 ABLATION STUDY

The ablation study reveals critical insights into the contributions of individual components (Planning without LoRA, Planning with LoRA, Control, Reflection only with stage 1, Reflection only with stage 2, Reflection with both stages) and Continual WM to LH tasks. We selected 10 random seeds for testing. Table 3 shows the mean and variance of the average success rate (SR) for each ablation study. When only PPO is used without any modules (first row), the agent fails to progress beyond basic tasks (28.16% SR for Wood, 0% for Iron+). Introducing the Planning module nearly doubles performance on Wood (45.69%) and marginally improves Stone (18.37%), but still fails to unlock advanced tasks (Iron+ at 0%), suggesting that planning alone cannot resolve the exploration bottleneck in LH tasks. A pivotal leap occurs when Control is added (P+C), with Wood and Stone success rates surging to 92.42% and 85.31%, respectively, and modest progress in Iron (31.59%). This underscores the necessity of structured exploration to navigate intermediate dependencies. However, the sharp decline in Gold (5.47%) and Diamond (3.52%) indicates persistent challenges in sparse reward scenarios. Integrating the Reflection module (P+C+R) achieves near-perfect Wood/Stone success (96.69%/93.82%) and significantly boosts Iron (42.61%), Gold (17.53%), and Diamond (10.09%), demonstrating its role in distilling exploration experiences to refine world models.

This experiment compares the results of Planning with LoRA and Planning without LoRA, demonstrating that LoRA training can significantly improve model convergence speed, reduce the number of invalid subtasks, and achieve autonomous task decomposition. We compare the effects of Reflection with stage 1 only, Reflection with stage 2 only, and Reflection with both stages. The experimental results show that combining task-based curriculum learning with data-based curriculum learning can significantly improve the model's average task success rate. EvoAgent is an improvement over DreamerV3. As shown in Figure 3, EvoAgent is significantly better than DreamerV3 because the CL-based reflector can greatly filter invalid exploration and accelerate model convergence.

As shown in Table 4, Table 5 and Table 6, for Eq. 18, Eq. 20 and Eq. 22, we perform an ablation experiment by running our agent in Minecraft with $10^7$ environment steps for the task "Iron". We report the average success rate (SR) on this task. The ablation experiment confirm that all metrics can improve model performance. As shown in Appendix B, compared to experience-free planners and text-only experience planners, the multimodal experience planner (EvoAgent) can maximize the planner's task decomposition capabilities.

## 5 CONCLUSION

This paper presents EvoAgent, a self-evolving agent with a continual World Model, which can autonomously complete various LH tasks across environments through self-planning, self-control, and self-reflection, without human intervention. EvoAgent contains three key modules: the memory-driven planner, the WM-guided action controller, and the experience-inspired reflector. In the future, we hope that our method can be truly applied to real robot scenarios.

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

# A  ONLINE MODEL-BASED REINFORCEMENT LEARNING (MBRL)

RL is typically formulated as a Markov Decision Process (MDP) defined by the tuple $(\mathcal{S}, \mathcal{A}, P, R, \gamma)$, where $\mathcal{S}$ is the state space, $\mathcal{A}$ is the action space, $P(s'|s, a)$ is the transition dynamics, $R(s, a)$ is the reward function, and $\gamma \in [0, 1)$ is the discount factor. The goal is to learn a policy $\pi(a|s)$ that maximizes the expected cumulative reward:

$$J(\pi) = \mathbb{E}_{\pi, P}\left[\sum_{t=0}^{\infty} \gamma^t R(s_t, a_t)\right], \tag{24}$$

In MBRL, the agent explicitly learns a model $\mathcal{M}$, which includes an approximate dynamics model $\hat{P}_\theta(s'|s, a)$ and a reward model $\hat{R}_\phi(s, a)$, parameterized by $\theta$ and $\phi$, respectively. These models are trained to minimize empirical prediction errors over observed transitions $\mathcal{D} = \{(s_i, a_i, s_i', r_i)\}$:

$$\mathcal{L}_{\text{model}}(\theta, \phi) = \mathbb{E}_{(s, a, s', r) \sim \mathcal{D}}\left[\|s' - \hat{P}_\theta(s, a)\|^2 + \|r - \hat{R}_\phi(s, a)\|^2\right], \tag{25}$$

Using the learned models, the agent performs planning to optimize its policy. For example, in value iteration, the state-value function $V(s)$ is iteratively updated via the Bellman equation:

$$V(s) \leftarrow \max_a [\hat{R}_\phi(s, a) + \gamma \mathbb{E}_{s' \sim \hat{P}_\theta(\cdot|s, a)} V(s')]. \tag{26}$$

In online MBRL, an agent interacts with the environment iteratively for $K$ rounds with the goal of learning a sequence to minimize $\mathcal{L}_{\text{model}}(\theta, \phi)$.

# B  MULTIMODAL PLANNER

Multimodal experience improves the planner over experience-free or text-only versions by providing grounded, perceptually-rich interaction history that enables context-aware subtask decomposition. Unlike an experience-free planner (which relies only on static prior knowledge) or a text-only version (which records symbolic descriptions), multimodal experience stores visual observations, self-states, and asset tokens. This allows the planner to learn from why subtasks succeed or fail in specific perceptual contexts, fine-tune via LoRA on actual visual-state sequences, and prioritize subtask sequences that have historically worked in similar multimodal settings.

Table 7: Performance comparison of experience-free planner, text-only experience planner, and multimodal experience planner.

|  | Experience-free Planner | Text-only Experience Planner | Multimodal Experience Planner |
|---|---|---|---|
| Iron | 2.58 | 6.34 | 51.07 |

As shown in the Table 7, we perform a comparative experiment by running our agent in Minecraft with $10^7$ environment steps for the task "Iron". We report the average success rate (SR) on this task. The experience-free planner: GPT-4o without finetune; the text-only experience planner: experience is textual information about whether the world model successfully executed subtasks; the multimodal experience planner: EvoAgent. The comparative experiment confirm that the multimodal experience can maximize the improvement of the planner.

## C  HYPERPARAMETERS

Table 8: EvoAgent hyperparameters.

| **General** | | |
| --- | :---: | :---: |
| Replay capacity | — | $5 \times 10^6$ |
| Batch size | $B$ | 16 |
| Batch length | $T$ | 64 |
| Activation | — | RMSNorm + SiLU |
| Learning rate | — | $4 \times 10^{-5}$ |
| Gradient clipping | — | AGC(0.3) |
| Optimizer | — | LaProp($\epsilon = 10^{-20}$) |
| **World Model** | | |
| Reconstruction loss scale | $\beta_{\text{pred}}$ | 1 |
| Dynamics loss scale | $\beta_{\text{dyn}}$ | 1 |
| Representation loss scale | $\beta_{\text{rep}}$ | 0.1 |
| Latent unimix | — | 1% |
| Free nats | — | 1 |
| **Actor Critic** | | |
| Imagination horizon | $H$ | 15 |
| Discount horizon | $1/(1-\gamma)$ | 333 |
| Return lambda | $\lambda$ | 0.95 |
| Critic loss scale | $\beta_{\text{val}}$ | 1 |
| Critic replay loss scale | $\beta_{\text{repval}}$ | 0.3 |
| Critic EMA regularizer | — | 1 |
| Critic EMA decay | — | 0.98 |
| Actor loss scale | $\beta_{\text{pol}}$ | 1 |
| Actor entropy regularizer | $\eta$ | $3 \times 10^{-4}$ |
| Actor unimix | — | 1% |
| Actor RetNorm scale | $S$ | $\text{Per}(R, 95) - \text{Per}(R, 5)$ |
| Actor RetNorm limit | $L$ | 1 |
| Actor RetNorm decay | — | 0.99 |
| **WM-Guided Action Controller** | | |
| Maximum episode step length | $T_{max}$ | 24000 |
| Task similarity threshold | $\sigma$ | 0.9 |
| Reward discount factor | $\gamma$ | 0.1 |
| **CL-based Reflector** | | |
| CL algorithm initialization threshold | $\rho_0$ | $5 \times 10^{-3}$ |
| CL subtask selection increase rate | $c_s$ | 0.3 |
| CL experience selection increase rate | $c_h$ | 0.5 |
| World model penalize weight | $\mu$ | 0.1 |

## D GENERALIZATION EXPERIMENT

**Atari100k.** The Atari100k benchmark is widely regarded as a key platform for testing data-efficient reinforcement learning methods. Unlike typical setups that permit agents to interact with environments for hundreds of millions of steps, Atari100k imposes a strict cap of 100k interactions (maximum episode length is 432K env steps), equivalent to about two hours of human play. This limited interaction budget forces algorithms to develop effective policies rapidly, rather than depending on massive-scale exploration or brute-force training. The benchmark spans 26 distinct games from the Arcade Learning Environment, encompassing challenges such as sparse reward signals, delayed credit assignment, and high-dimensional pixel inputs. Results are usually reported using normalized human scores, ensuring comparability across games with diverse dynamics. By constraining available data so severely, Atari100k serves as a rigorous probe into the adaptability of reinforcement learning systems, offering insights into the generalization capacity of model-based approaches, world modeling strategies, and representation learning techniques.

**Baselines.** Random indicates that each action decision is randomly selected. Human refers to collecting video recordings of humans playing the game and calculating the average score. PPO (Schulman et al., 2017) is a classic model-free reinforcement learning algorithm, and dreamverV3 (Hafner et al., 2025) is a classic model-based reinforcement learning algorithm.

**Experimental Settings.** We adopt the same experimental settings as Dreamerv3. Except for the EvoAgent experimental results, the rest are the publicly available experimental results of Dreamerv3. The experimental results are shown in the Table 9.

Table 9: Atari100k scores.

| Task | Random | Human | PPO | DreamerV3 | EvoAgent |
|---|---|---|---|---|---|
| Steps | — | — | 400K | 400K | 400K |
| Alien | 228 | 7128 | 276 | 1118 | **1392** |
| Amidar | 6 | 1720 | 26 | 97 | **329** |
| Assault | 222 | 742 | 327 | 683 | **981** |
| Asterix | 210 | 8503 | 292 | 1062 | **1492** |
| Bank Heist | 14 | 753 | 14 | **398** | 362 |
| Battle Zone | 2360 | 37188 | 2233 | 20300 | **24830** |
| Boxing | 0 | 12 | 3 | 82 | **91** |
| Breakout | 2 | 30 | 3 | 10 | **13** |
| Chopper Command | 811 | 7388 | 1005 | 2222 | **4375** |
| Crazy Climber | 10780 | 35829 | 14675 | **86225** | 78215 |
| Demon Attack | 152 | 1971 | 160 | 577 | **1205** |
| Freeway | 0 | 30 | 2 | 0 | **5** |
| Frostbite | 65 | 4335 | 127 | 3377 | **3674** |
| Gopher | 258 | 2412 | 368 | 2160 | **2219** |
| Hero | 1027 | 30826 | 2596 | **13354** | 12168 |
| Jamesbond | 29 | 303 | 41 | 540 | **621** |
| Kangaroo | 52 | 3035 | 55 | 2643 | **2753** |
| Krull | 1598 | 2666 | 3222 | 8171 | **10027** |
| Kung Fu Master | 258 | 22736 | 2090 | 25900 | **28692** |
| Ms Pacman | 307 | 6952 | 366 | 1521 | **3246** |
| Pong | -21 | 15 | -20 | -4 | **-2** |
| Private Eye | 15 | 69571 | 100 | 3238 | **5285** |
| Qbert | 164 | 13455 | 317 | 2921 | **4793** |
| Road Runner | 12 | 7845 | 602 | 19230 | **21703** |
| Seaquest | 68 | 42055 | 305 | 962 | **2305** |
| Up N Down | 533 | 11693 | 1502 | **46910** | 37284 |

**Experimental results analysis.** As shown in Table 9, the experimental results highlight that EvoAgent achieves superior generalization across the Atari100k suite, outperforming existing methods and achieving human-level performance in some tasks. This advantage arises from the continual world model combined with its iterative planning–control–reflection cycle, enabling the agent to avoid narrow overfitting and adapt to novel task dynamics.

