# OpenReview forum: "EvoAgent: Self-evolving Agent with Continual World Model for Long-Horizon Tasks"
_ICLR.cc/2026/Conference — Submitted to ICLR 2026_

### Official Review · Reviewer_Khpi · 2025-10-30

**Soundness:** 4
**Presentation:** 4
**Contribution:** 4
**Rating:** 4
**Confidence:** 4

**Summary:**

The paper proposes **EvoAgent**, a self-evolving agent with a **Continual World Model**, which addresses the challenges of **experience dependency and catastrophic forgetting** in long-horizon tasks. By enabling **self-planning, self-control, and self-reflection**, EvoAgent autonomously decomposes tasks and updates world knowledge. In experiments on **Minecraft** and **Atari**, it achieved an **average success rate improvement of 105%** and **reduced ineffective actions by over six times**.

**Strengths:**

1. **The Novel Self-evolving Agent Framework**
    The paper introduces **EvoAgent**, a self-evolving agent with a **Continual World Model (CWM)** that enables autonomous *self-planning, self-control,* and *self-reflection* without human intervention. The proposed *planning–control–reflection* loop forms a coherent self-improving mechanism, addressing the limitation of one-directional learning in prior agents.
2. **Effective Continual Learning and Forgetting Mitigation**
    **EvoAgent** achieves continual world knowledge updates and prevents catastrophic forgetting through weighted experience replay, Fisher-based regularization, and LoRA fine-tuning for adaptive task learning.
3. **Strong Technical Integration**
    The framework seamlessly combines **LLMs** for task decomposition, **World Models (RSSM)** for control, and **Curriculum Learning** for experience selection—enhancing autonomy and generalization.
4. **Comprehensive and Convincing Experiments**
    On **Minecraft** and **Atari**, EvoAgent improves average success rates by **105%** and reduces ineffective actions **sixfold**. Ablation studies show the continual world model contributes **72%** of the performance gain.
5. **Clear Structure and Logical Presentation**
    The paper is well-organized, with a coherent narrative from motivation to validation, making its technical and empirical contributions easy to follow.

**Weaknesses:**

1. **Typographical  issues.**
    For example, the paper states “including WM-based agents (such as PPO Schulman et al. (2017), DreamerV3 Hafner et al. (2025)),” but PPO is a **model-free** algorithm rather than a WM-based method.
2. **Lack of quantitative evidence for the “6× fewer ineffective actions” claim.**
    The abstract mentions this improvement, but no explicit numerical data or training performance curves are provided to support it—only indirect reflection via the final EE metric. Adding corresponding training curves would greatly strengthen this claim’s credibility.
3. **Insufficient details on the reward predictor.**
    Unlike DreamerV3, EvoAgent’s reward predictor is goal-conditioned. However, the paper does not specify its structure and training objective.
4. **Unclear visual encoding path in the Experience-driven Task Planner.**
    Since the experience pool $D_{MEP}$ includes visual observations (Obs), these should logically pass through an encoder and projector. Yet this process is not clearly illustrated or described, which may cause confusion about the data flow.
5. **Incomplete experimental comparison.**
     The experimental section only compares EvoAgent against PPO, DreamerV3, Jarvis-1, and Optimus-1, but omits **more recent model-based baselines**, such as *LS-Imagine* [1].
6. **Unreported computational cost of GPT-4o usage.**
    The paper employs GPT-4o for online learning, but the token consumption and computational cost are not reported. It would be helpful to clarify whether this process is computationally expensive or practical for large-scale applications.

[1] Wang Qi, er al. Open-World Reinforcement Learning over Long Short-Term Imagination,  **ICLR 2025 Oral**

**Questions:**

1. **On text tokenization of multimodal data.**
    How can (D_{exp})—which contains actions and structured numerical information—be directly fed into a text tokenizer? Can GPT-4o truly interpret such structured multimodal data, or is there an intermediate formatting or symbolic conversion process?
2. **Alternative world models.**
    Have the authors experimented with different world model architectures, such as **Transformer-based approaches (e.g., DiT or Decision Transformer)**? If not, what motivated the choice of RSSM?
3. **Transferability to robotics.**
    Could EvoAgent be extended to real or simulated robotic settings (e.g., **Libero** or manipulation tasks)?

---

> ### Author Response · Authors · 2025-12-03
> **Rseponse to Reviewer Khpi (Part 1 of 2)**
>
> Thank you very much for your detailed review and valuable feedback. We have incorporated your suggestions, particularly by including quantitative analysis and comparative experiments, and we hope these address your concerns.
>
> This response is a little delayed due to the substantial time required for the additional experiments. Thank you for your understanding :)
>
> ---
>
> **W1: Typographical issues.**
>
> **Answer:** We have reviewed the typographical issues in full paper and updated them in the revised PDF version.
>
> ---
>
> **W2:  Lack of quantitative evidence for the “6× fewer ineffective actions” claim.**
>
> **Answer:** As shown in Table 1, EvoAgent improves the average exploration efficiency by 603.28% compared to existing algorithms Jarvis-1, dreamerV3, LS-Imagine, and Optimus-1, achieving "6 × fewer ineffective actions".
>
> |  Group  | Metric | PPO  | GPT-4V | Jarvis-1 | dreamerV3 | **LS-Imagine** | Optimus-1 | **EvoAgent** |          **Improving** (%)          |
> | :-----: | :----: | :--: | :----: | :------: | :-------: | :------------: | :-------: | :----------: | :---------------------------------: |
> | Diamond |  SR↑   | 0.00 |  0.00  |   7.69   |   4.73    |      4.36      |   9.30    |  **17.36**   |               166.26                |
> | Diamond |  EE↑   | 0.00 |  0.00  |   0.07   |   3.69    |      4.19      |   7.31    |  **26.83**   | **603.28** |
>
> ---
>
> **W3: Insufficient details on the reward predictor.**
>
> **Answer:** Thank you for the insightful comment. EvoAgent's goal-conditioned reward predictor ($\hat{r}\_t \sim r\_\theta(\hat{r}\_t | h\_t,z\_t, \text{Emb}{g\_i})$) is an extension of the DreamerV3 predictor ($\hat{r}\_t \sim r\_\theta(\hat{r}\_t | h\_t,z\_t)$). As shown in Eq. 14, the reward $R(s\_t, a\_t, g\_i)$ considers not only the deterministic latent state $h_t$ and stochastic latent variable $z_t$ based on the current observation states $s\_t$ and actions $a\_t$ but also a goal embedding $\text{Emb}{g\_i}$ derived from the current subtask $g_i$. The predictor is implemented as a multi-layer perceptron (MLP) that outputs a scalar value $\hat{R}\_t$. Its training objective is to minimize the mean squared error (MSE) between the predicted reward $\hat{R}t$ and the actual environmental reward $R_t$. This loss is part of the overall world model loss $\mathcal{L}{\text{pred}}$ used for updating $\mathcal{M}_w$ (Eq. 22). We have updated this part at Lines 255-258 in the revised PDF.
>
> ---
>
> **W4-Q1: Unclear visual encoding path in the Experience-driven Task Planner. Text tokenization of multimodal data.**
>
> **Answer:** $\mathcal{D}\_{\text{MEP}}$ includes both image and text data, with image data also processed through a visual encoder and projector. As shown in Figure 2, we adopt the image tokenizer $f\_{v}$ to encode the raw images $\mathcal{O}\_{obs}, \mathcal{S}\_{\text{self}}, \mathcal{S}\_{\text{assets}}$ into token embeddings $\mathcal{V}=\\{v_1,v_2,...,v_n\\} \in \mathbb{R}^{n \times d}$, where $n$ denotes the number of visual tokens and $d$ is the dimensionality of each token. We adopt the textual tokenizer $f\_{t}$ to encode $\mathcal{T}$ into token embeddings. We further utilize a lightweight projection module $f\_l$ with a trainable projection matrix $W$. This module maps the visual tokens to the same space with text embeddings $\hat{\mathcal{V}} = W \mathcal{V}$, yielding $\hat{\mathcal{V}}=\{\hat{v}\_1,\hat{v}\_2,...,\hat{v}\_n\} \in \mathbb{R}^{n \times d}$. The output of our planner is the subtask $g\_i$. We have updated this part at Lines 221-226 in the revised PDF.

---

> ### Author Response · Authors · 2025-12-03
> **Rseponse to Reviewer Khpi (Part 2 of 2)**
>
> **W5: Incomplete experimental comparison**
>
> **Answer:** Thank you for the insightful comment. We have added LS-Imagine as a WM-based baseline. The experimental results are shown in the table below.
>
> |  Group  | Metric | dreamerV3 | LS-Imagine | EvoAgent  |
> | :-----: | :----: | :-------: | :--------: | :-------: |
> |  Wood   |  SR↑   |   91.07   |   95.87    | **97.47** |
> |  Wood   |  EE↑   |   93.22   |   97.41    | **98.43** |
> |  Stone  |  SR↑   |   86.82   |    91.5    | **94.53** |
> |  Stone  |  EE↑   |   88.39   |   92.36    | **96.48** |
> |  Iron   |  SR↑   |   33.79   |   35.82    | **51.82** |
> |  Iron   |  EE↑   |   35.68   |   38.27    | **58.54** |
> |  Gold   |  SR↑   |   6.57    |    6.61    | **21.69** |
> |  Gold   |  EE↑   |   8.05    |   10.69    | **30.48** |
> | Diamond |  SR↑   |   4.73    |    4.36    | **17.36** |
> | Diamond |  EE↑   |   3.69    |    4.19    | **26.83** |
> | Overall |  SR↑   |   15.03   |    15.6    | **30.29** |
>
> LS-Imagine, through a hybrid approach of short-term and long-term imagination, significantly outperforms DreamerV3 in SR and EE metrics on the basic Wood/Stone task. However, its performance growth is slow on complex tasks because accumulated errors can mislead the optimization direction. EvoAgent with a continual world model can  address this problem through closed-loop dynamic self-planning, self-control, and self-reflection, autonomously updating the multimodal experience pool and world knowledge, filtering invalid exploration, mitigating historical forgetting, and contributing to better autonomous planning and action control. We have added this comparative experiment at Table 1 and Lines 483-485 in the revised PDF.
>
> ---
>
> **W6: The computational cost of GPT-4o usage**
>
> **Answer:** LLM API calls occur in two processes: subtask planning and subtask failure fine-tuning. The maximum execution steps of each subtask is $T_{max}=24000$ (Hafner et al., 2025). As the agent self-evolves, the number of subtask failures decreases, which greatly reduces the overhead of subtask failure fine-tuning. Throughout the experiment, with an average of 750 planning calls over $10^7$ environment steps, we spent about $\\$90$ to access the GPT-4o API. We have already added the cost of LLM API calls at Lines 420-424 in the revised PDF.
>
> ---
>
> **Q2: Alternative world models.**
>
> **Answer:** EvoAgent adopts RSSM over Transformer-based architectures primarily for its superior handling of partial observability, sequential latent state modeling, and online continual learning in dynamic environments. RSSM explicitly models stochastic latent transitions and deterministic recurrent states, enabling robust long-horizon prediction and stable knowledge integration without catastrophic forgetting. While Transformers excel in pattern recognition, they are less sample-efficient for continual world model updates and struggle with coherent long-horizon state prediction under partial observability, making RSSM more suitable for EvoAgent’s closed-loop dynamic and self-evolution objectives.
>
> ---
>
> **Q3: Transferability to robotics.**
>
> **Answer:** Yes, EvoAgent is highly suitable for extension to real or simulated robotic settings such as the Libero benchmark or general manipulation tasks. EvoAgent with a continual world model can directly address key robotic challenges: long-horizon planning under partial observability, online adaptation to new objects or environments, and mitigation of catastrophic forgetting across sequential tasks. The system’s ability to autonomously decompose tasks, learn from multimodal experience, and self-verify progress aligns well with the demands of robotic manipulation, where tool use, multi-stage goals, and dynamic conditions are common. This paper demonstrates through experiments that EvoAgent not only achieves optimal performance in complex Minecraft environments, but also maintains optimal performance in the Atair game, indicating that EvoAgent has extremely strong transferability.
>
> ---
>
> We sincerely thank you for your valuable and constructive feedback. We will integrate these insightful comments into our revised revision. Unfortunately, the current rating puts us in a difficult position. We sincerely hope our responses are sufficient to address your concerns. We would be very grateful for your timely re-evaluation, as we would strongly prefer to continue with this submission.

---

### Official Review · Reviewer_ZkBX · 2025-10-31

**Soundness:** 3
**Presentation:** 2
**Contribution:** 2
**Rating:** 4
**Confidence:** 3

**Summary:**

This paper presents EvoAgent, a novel framework designed to address the challenges of long-horizon (LH) task completion in open-ended environments. The core contribution is a self-evolving agent that integrates a continual World Model (WM) with three key modules for planning, control, and reflection, enabling autonomous experience accumulation and knowledge updates without human intervention. The paper is well-structured, tackles a significant and timely problem in embodied AI, and is supported by extensive experiments in Minecraft and Atari, showing impressive quantitative improvements over strong baselines.

**Strengths:**

The proposed integration of a continual WM within a closed-loop planning-control-reflection cycle is a compelling and timely contribution. Addressing the limitations of existing methods (reliance on human curricula, catastrophic forgetting) by enabling agents to autonomously update and select multimodal experiences is a significant step forward for long-horizon task solving.

The experimental evaluation is thorough and convincing. The use of the challenging Minecraft benchmark provides strong evidence for the method's effectiveness and generalization capability.

**Weaknesses:**

1. The paper's writing needs to be improved a lot. For example, almost all the citations are not appropriately shown (without the parenthesis), which makes the understanding difficult; On line 207, the model-based RL is introduced, while it seems no format text refers to it. For action selection, there is no technical details provided about how to solve Eq 14. On the other hand, the basic machnism of LLM (line 247-253) can be omited to save the space.

2. The core contributions needs to be more clear. There are too many new modules within EvoAgent, including the Task Planner, the Action Controller, the continual World Model, and the Reflector. Are all these modules new? If yes, their novelties should be made more clear, with necessary ablation experiments need to be include, for example:
- why there must be two stages of CL
- why there must be some many terms on Eq 18,  20 and 22
- why these must be a self-verification for controller.

(Considering the page limit of a conference paper, it is suggested to narrow the topic such that the core contribution can be more clear.)
On the other hand, if the answer is no, there should be more content moved to preliminary and the author can focus to interpretate the novelty of the rest part.

3. Why use only the failure expeirence to LoRA finetune the planner? Why the multimodal experience can improve the planner? How the mechanism  altered compared to the experience-free version, or with text-only experience? There should be more in-depth analysis in the experiment.

**Questions:**

- How is the computational overhead of the full EvoAgent loop (especially the two-stage CL and LLM fine-tuning) compared to a baseline like DreamerV3?
- why there must be two stages of CL
- why these must be a self-verification for controller.
- why use only the failure expeirence to LoRA finetune the planner
- why the multimodal experience can improve the planner?

---

> ### Author Response · Authors · 2025-12-03
> **Rseponse to Reviewer ZkBX (Part 1 of 3)**
>
> Thank you very much for your detailed review and valuable feedback. We have incorporated your suggestions, particularly by including additional sensitivity analysis and computational overhead analysis, and we hope these address your concerns.
>
> This response is a little delayed due to the substantial time required for the additional experiments. Thank you for your understanding :)
>
> ---
>
> **W1: The paper's writing needs to be improved a lot.**
>
> **Answer:** We thank the reviewer for these constructive writing suggestions.
>
> - **Without the parentheses**: We have corrected all citation formats to include parentheses for clarity.
> - **Where use Model-based RL:** As shown in the Introduction, "By utilizing a model-based online RL setup and closed-loop dynamics, EvoAgent is able to autonomously update the multimodal experience pool and world knowledge". EvoAgent has a model-based online RL setup, and I have already made this clear at Lines 142-143 in the revised PDF.
> - **How to solve Eq 14:** We have added technical details at Lines 258-262 in the revised PDF. At each time step $t$, we sample a population of $N$ action sequences $\\{a\_{t:t+H}^{(k)}\\}\_{k=1}^{N}$ from the action space $\mathcal{A}^H$. The world model $\mathcal{M}\_{w}$ is used to predict the future states and compute the expected cumulative reward $\mathbb{E}\_{\mathcal{M}\_{w}}\left[\sum\_{\tau=t}^{t+H} \gamma^{\tau-t} R(s\_{\tau}, a\_{\tau}, g\_i)\right]$ for each sequence. The sequence $a_{t:t+H}^{\*}$ with the highest rewards is selected, and the first action $a_t^{\*}$ of this sequence is executed in the environment.
> - **The basic mechanism of LLM (Lines 247-253) can be omitted:** Thank you very much for your suggestion. We have removed this part.
>
> ---
>
> **W2: The core contributions need to be clearer.**
>
> **Answer:** Thank you very much for your recognition of this work. As shown in Eq. 2 and Figure 2, these modules are all new and essential components of self-evolving agents. In Section 3 of the revised PDF, we highlighted the contribution of each module and verified it through ablation experiments.
>
> - **The experience-driven task planner** uses an LLM along with interaction experiences to incorporate self-state into the planning phase and convert LH tasks into executable sub-tasks. **The core contribution** is how agents can update the LLM for efficient long-horizon task planning without human intervention as the agent interacts dynamically with the environment. The solution is highlighted at Lines 211-242 in the revised PDF, as shown in the answer to W3-Q4.
> - **The WM-guided action controller** leverages WM to generate low-level actions and incorporates a self-verification mechanism to update multimodal experiences. **The core contribution** is how agents can reduce inefficient exploration and achieve efficient task execution without human intervention. The solution is highlighted at Lines 246-293 in the revised PDF, as shown in the answer to Q3.
> - **The Curriculum Learning (CL) -based reflector** implements a two-stage CL algorithm to select experiences for task-adaptive WM updates. **The core contribution** is how agents can efficiently update the world model without human intervention as the agent interacts dynamically with the environment. The solution is highlighted at Lines 298-365 in the revised PDF, as shown in the answer to Q2.
> - **The continual world model**, which can connect the three separate modules: the task planner, the action controller and the reflector. Agent with a continual world model can self-evolve **through closed-loop dynamic** self-planning, self-control, and self-reflection, autonomously updating the multimodal experience pool and world knowledge. The solution is also highlighted in the revised PDF, as shown in Figure 2 and Algorithm 1.

---

> ### Author Response · Authors · 2025-12-03
> **Rseponse to Reviewer ZkBX (Part 2 of 3)**
>
> **W2: Necessary ablation experiments need to be included.**
>
> **Answer:** As shown in Table 3, we provide detailed ablation experiments. More ablation experiments are shown below.
>
> - **Q2: Why must there be two stages of CL.** The two-stage CL is crucial because it allows for more efficient updates to the world model at both the macro (subtask selection) and micro (experience selection) levels, reducing the impact of invalid data on model performance and reducing computational costs. Using only one stage is insufficient: selecting subtasks without filtering internal experiences still results in **low-quality data hindering learning**; while filtering experiences without task-level selection **increases computational costs** since all experiences require weight calculation. As shown in Table 3, ablation experiments confirm that both stages are necessary for optimal performance because they together achieve efficient world model updates. We select 10 random seeds for testing and report the average success rate (SR) on each task group. R.$^{1}$, R.$^{2}$, and R. represent Reflection only with stage 1, Reflection only with stage 2, and Reflection with both stages, respectively. The PPO algorithm is used by default for model decision-making.
>
>   |  R¹  |  R²  |  R   |    Wood    |   Stone    |    Iron    |    Gold    |  Diamond   |
>   | :--: | :--: | :--: | :--------: | :--------: | :--------: | :--------: | :--------: |
>   |  ✓   |      |      | 93.18±2.86 | 87.72±5.74 | 34.63±5.19 | 8.68±2.72  | 5.14±2.51  |
>   |      |  ✓   |      | 95.37±2.48 | 91.26±3.86 | 39.58±5.08 | 14.20±4.05 | 8.93±3.73  |
>   |      |      |  ✓   | 96.69±2.24 | 93.82±3.34 | 42.61±4.80 | 17.53±5.18 | 10.09±3.54 |
>
> - **Why there must be many terms on Eq. 18, 20 and 22.** As shown in the Table below, for the four metrics in Eq. 18: Relevance (R.), Efficiency (E.), Importance (I.), and Completion Rate (C.R.), we perform an ablation experiment by running our agent in Minecraft with $10^7$ environment steps for the task "Iron". We report the average success rate (SR) on this task. The ablation experiment confirms that all metrics can improve model performance. This ablation experiment has been added at Lines 487-498 in the revised PDF.
>
>   |  R.  |  E.  |  I.  | C.R. | Iron  |
>   | :--: | :--: | :--: | :--: | :---: |
>   |  ✓   |      |      |      | 40.16 |
>   |  ✓   |  ✓   |      |      | 41.77 |
>   |  ✓   |  ✓   |  ✓   |      | 46.54 |
>   |  ✓   |  ✓   |  ✓   |  ✓   | 49.37 |
>
>   As shown in the Table below, for the three metrics in Eq. 20: TD-Error (TD-R.),  Gradient Norm (G.N.), and  Information Gain (I.G.), we perform an ablation experiment by running our agent in Minecraft with $10^7$ environment steps for the task "Iron". We report the average success rate (SR) on this task. The ablation experiment confirms that all metrics can improve model performance. This ablation experiment has been added at Lines 487-498 in the revised PDF.
>
>   | TD-R. | G.N. | I.G. | Iron  |
>   | :---: | :--: | :--: | :---: |
>   |   ✓   |      |      | 42.72 |
>   |   ✓   |  ✓   |      | 45.59 |
>   |   ✓   |  ✓   |  ✓   | 50.43 |
>
>   As shown in the Table below, for the three metrics in Eq. 22: Curriculum Loss and Regularization, we perform an ablation experiment by running our agent in Minecraft with $10^7$ environment steps for the task "Iron". We report the average success rate (SR) on this task. The ablation experiment confirms that all metrics can improve model performance. This ablation experiment has been added at Lines 487-498 in the revised PDF.
>
>   | Curriculum Loss | Regularization | Iron  |
>   | :-------------: | :------------: | :---: |
>   |        ✓        |                | 48.61 |
>   |        ✓        |       ✓        | 50.92 |
>
> - **Q3: Why these must be a self-verification for controller.** Since the task controller is task-driven, the self-verification mechanism is used to determine whether the subtask $g\_i$ can be terminated. It can greatly reduce ineffective exploration. As shown in the Table below, we performed a sensitivity analysis by running our agent in Minecraft with $10^7$ environment steps for the task "Iron". Experimental results show that the task success rate remains stable when $\sigma \in \[0.875, 0.925\]$, with sharp declines outside this range due to over/under-termination. When $\sigma \textless 0.875$, sub-tasks may not be completed but are misjudged, causing subsequent tasks to fail. When $\sigma \textgreater 0.925$, due to strict self-verification, sub-tasks may be completed but still require re-planning, reducing the task completion rate.
>
>   | $\sigma$          | 0.8  | 0.825 | 0.85  | 0.875 | 0.9   | 0.925 | 0.95  | 0.975 |
>   | ----------------- | ---- | ----- | ----- | ----- | ----- | ----- | ----- | ----- |
>   | Success Rate (SR) | 5.32 | 17.03 | 34.48 | 48.69 | 52.43 | 47.72 | 26.74 | 3.64  |
>
> ---

---

> ### Author Response · Authors · 2025-12-03
> **Rseponse to Reviewer ZkBX (Part 3 of 3)**
>
> **W3-Q4: Why use only the failure expeirence to LoRA finetune the planner?**
>
> **Answer:** As shown in Subsection 3.2, the planner update includes 3 phases:
>
> 1. During agent initialization, the fine-tuning process utilizes all accumulated experiences from the multimodal experience pool $\mathcal{D}_{\text{MEP}}$ for task planning. When the multimodal experience pool is empty, the agent initializes task planning based on the capabilities of the original GPT-4o.
> 2. During the agent's lifecycle, when the WM-guided action controller feedback indicates the subtask $g_i$ failure, experience trajectories relevant to the subtask $g_i$ by label matching are extracted to construct input-output pairs $\{(X_{\text{in}}^{(k)}, Y_{\text{out}}^{(k)})\}$ for model fine-tuning, where the input $X_{\text{in}}^{(k)}$ includes all the experience $h$ related the subtask $g_i$, while the output $Y_{\text{out}}^{(k)}$ represents the corresponding subtask sequence. The optimization objective is to maximize the cross-entropy loss between the predicted and history subtask sequences. This enables the planner to quickly study from the failure patterns while preserving its general planning capabilities, thereby improving robustness and reducing repeated errors in long-horizon tasks.
> 3. When the agent dies (health value is 0), the agent is reinitialized.
>
> ---
>
> **W3-Q5: Why can the multimodal experience improve the planner? How does the mechanism alter compared to the experience-free planner, or with text-only experience planner? There should be a more in-depth analysis in the experiment.**
>
> **Answer:** Multimodal experience improves the planner over experience-free or text-only versions by providing grounded, perceptually-rich interaction history that enables context-aware subtask decomposition. Unlike an experience-free planner (which relies only on static prior knowledge) or a text-only version (which records symbolic descriptions), multimodal experience stores visual observations, self-states, and asset tokens. This allows the planner to learn from why subtasks succeed or fail in specific perceptual contexts, fine-tune via LoRA on actual visual-state sequences, and prioritize subtask sequences that have historically worked in similar multimodal settings.
>
> As shown in the Table below, we perform a comparative experiment by running our agent in Minecraft with $10^7$ environment steps for the task "Iron". We report the average success rate (SR) on this task. **The experience-free planner**: GPT-4o without finetune; **the text-only experience planner**: experience is textual information about whether the world model successfully executed subtasks; **the multimodal experience planner**: EvoAgent. The comparative experiment confirms that the multimodal experience can maximize the improvement of the planner. This comparative experiment has been added at Lines 727-745 in the revised PDF.
>
> |      | Experience-free Planner | Text-only Experience Planner | Multimodal Experience Planner (EvoAgent) |
> | :--: | :---------------------: | :--------------------------: | :--------------------------------------: |
> | Iron |          2.58           |             6.34             |                  51.07                   |
>
>
> ---
>
>
> **Q1: How is the computational overhead of the full EvoAgent loop compared to a baseline like DreamerV3?**
>
> **Answer:**  As shown in Lines 404-406, EvoAgent and Dreamerv3 runs on a single A100 GPU. Taking $10^7$ steps as an example, compared to dreamerV3 running for 7 days, EvoAgent only needs to run for 2.7 days.
>
> ---
>
> We sincerely thank you for your valuable and constructive feedback. We will integrate these insightful comments into our revised revision. Unfortunately, the current rating puts us in a difficult position. We sincerely hope our responses are sufficient to address your concerns. We would be very grateful for your timely re-evaluation, as we would strongly prefer to continue with this submission.

---

### Official Review · Reviewer_d7no · 2025-11-02

**Soundness:** 3
**Presentation:** 2
**Contribution:** 3
**Rating:** 6
**Confidence:** 3

**Summary:**

This paper proposes EvoAgent, a self-evolving agent for long-horizon tasks that combines an LLM-based planner, RSSM-based world model controller, and a reflector that updates the world model continually by a hierarchical trajectory selection from the Multi-model Experience Pool (MEP). The selection is adaptively based on relevance, efficiency, importance, and complete ratio. The key contribution is a continual world model that autonomously updates through closed-loop planning-control-reflection, addressing catastrophic forgetting in sequential task learning. Experiments are conducted on Atari and Minecraft, showing an average 105% improvement in success rate over baselines on the latter setting.

**Strengths:**

- Important problem formulation: The paper addresses autonomous experience accumulation and world knowledge updates for long-horizon tasks, eliminating reliance on human-designed curricula or demonstrations, which is critical for real-world deployment.
- Novel continual world model design: The two-stage curriculum learning using relevance, efficiency, world model divergence, TD-error, and gradient norms as weight signals is innovative, prioritizing experiences that maximally update environmental understanding efficiently.
- Strong empirical results: Achieves 105% average success rate improvement and 6× efficiency gains over strong baselines, with performance advantages increasing systematically on harder, longer-horizon tasks.
- Sound experimental design: Tests across Minecraft (67 tasks) and Atari environments, compares against diverse baselines (model-free, model-based, LLM-based), uses appropriate metrics (success rate, exploration efficiency).

**Weaknesses:**

- Minor spelling errors: "Atari" is spelled "Atair" in multiple instances.
- Some ambiguities in method description:
  - No explicit description on how to determine "relevant" experiences from MEP for planner fine-tuning.
  - Figure 2 is daunting and unintuitive, without clear temporal pipeline of in what order each module infers or updates.
- Missing analysis on cost of LLM API calls. This is significant because the method relies on LLM inference for frequent planning.
- Some personal concerns on the soundness of the method. Please see questions.

**Questions:**

- Section 3.2 states "experience trajectories relevant to the subtask $g_i$ are extracted" for LoRA fine-tuning after failures, but never specifies how relevance is determined. Is it exact label matching, embedding-based similarity, or another method?
- The fine-tuning uses pairs $\{(X_{in}^{(k)}, X_{out}^{(k)})\}$ where inputs are lower-level experiences and outputs are higher-level subtasks. Both are denoted "X" despite representing different modalities, making the supervision structure unclear. Can you clarify this notation and confirm the understanding?
- Subtask failures trigger planner fine-tuning, but failures can stem from poor world models, suboptimal control, or environmental stochasticity, not just planning errors. Why should the planner always update on failures? Moreover, fine-tuning requires successful subtask sequences as supervision. How does this work initially when MEP contains no successful experiences?
- The self-verification threshold σ=0.9 is labeled "set empirically" without principled analysis. Is there sensitivity analysis, validation-based tuning, or theoretical justification? How are state embeddings from WM and goal embeddings from LLM aligned for meaningful cosine similarity?

---

> ### Author Response · Authors · 2025-12-03
> **Rseponse to Reviewer d7no (Part 1 of 2)**
>
> Thank you very much for your detailed review and valuable feedback. We have incorporated your suggestions, particularly by including the cost analysis of LLM API calls and the sensitivity analysis of parameters, and we hope these address your concerns.
>
> This response is a little delayed due to the substantial time required for the additional experiments. Thank you for your understanding :)
>
> ---
>
> **W1: Minor spelling errors**
>
> **Answer:** We have carefully checked all spelling errors and updated them in the revised PDF.
>
> ---
>
> **W2.1-Q1:  No explicit description on how to determine "relevant" experiences from MEP for planner fine-tuning. Is it exact label matching, embedding-based similarity, or another method?**
>
> **Answer:** As shown in Eq. 12:  $\mathcal{D}\_{\text{MEP}} = \\{h\\}, h = \langle (s\_t, a\_t, r\_t, s_{t+1}), \mathbb{P}\_{(g\_i)} | g\_i \rangle$, where $h$ represents the experience; $s_t$ represents multimodal state at step $t$; $r_t$ represents the reward obtained by performing action $a_t$ at state $s_t$; $g_i$ is the subtask; $\mathbb{P}(g_i)$ indicates the percentage of subtask $g_i$ completion.
>
> Each experience $h$ is task-specific, so experience trajectories relevant to the subtask $g_i$ are extracted by label matching. In the revised PDF, we have updated this explicit description at Lines 234-237.
>
> ---
>
> **W2.2: Figure 2 is daunting and unintuitive, without clear temporal pipeline of in what order each module infers or updates.**
>
> **Answer:** Your suggestion is excellent, as shown in Eq. 2, the sketch of EvoAgent illustrates the clear temporal pipeline $\text{Planner} \rightarrow \text{Controller} \rightarrow \text{Reflector} \rightarrow$. We have modified Figure 2 based on this sketch to clarify the execution order of each module.
>
> ---
>
> **W3: Missing analysis on the cost of LLM API calls**
>
> **Answer:** LLM API calls occur in two processes: subtask planning and subtask failure fine-tuning. The maximum execution steps of each subtask is $T_{max}=24000$ (Hafner et al., 2025). As the agent self-evolves, the number of subtask failures decreases, which greatly reduces the overhead of subtask failure fine-tuning. Throughout the experiment, with an average of 750 planning calls over $10^7$ environment steps, we spent about $\\$90$ to access the GPT-4o API. We have already added the cost of LLM API calls at Lines 420-424 in the revised PDF.
>
> ---
> **Q2: Please clarify the notation "X"**
>
> **Answer:** Your understanding is correct. The fine-tuning uses pairs ${(X_{in}^{(k)}, X_{out}^{(k)})}$ where inputs are lower-level experiences and outputs are higher-level subtasks. $X_{in}^{(k)}$ and $X_{out}^{(k)}$ represent different modalities. To address the issue that "Both are denoted 'X' despite representing different modalities, making the supervision structure unclear," we modified it to ${(X_{in}^{(k)}, Y_{out}^{(k)})}$, and explained this at Lines 234-237 in the revised PDF.

---

> ### Author Response · Authors · 2025-12-03
> **Rseponse to Reviewer d7no (Part 2 of 2)**
>
> **Q3: Why should the planner always update on failures? How does this work initially when MEP contains no successful experiences?**
>
> **Answer:** As shown in Subsection 3.2, the planner update includes 3 phases:
>
> 1. During agent initialization, the fine-tuning process utilizes all accumulated experiences from the multimodal experience pool $\mathcal{D}_{\text{MEP}}$ for task planning.  When the multimodal experience pool is empty, the agent initializes task planning based on the capabilities of the original GPT-4o.
>
> 2. During the agent's lifecycle, when the WM-guided action controller feedback indicates the subtask $g_i$ failure, experience trajectories relevant to the subtask $g_i$ by label matching are extracted to construct input-output pairs $\{(X_{\text{in}}^{(k)}, Y_{\text{out}}^{(k)})\}$ for model fine-tuning, where the input $X_{\text{in}}^{(k)}$ includes all the experience $h$ related the subtask $g_i$, while the output $Y_{\text{out}}^{(k)}$ represents the corresponding subtask sequence. The optimization objective is to maximize the cross-entropy loss between the predicted and history subtask sequences. This enables the planner to quickly study from the failure patterns while preserving its general planning capabilities, thereby improving robustness and reducing repeated errors in long-horizon tasks.
>
> 3. When the agent dies (health value is 0), the agent is reinitialized.
>
> We apologize for any confusion caused by our unclear wording. When MEP contains no successful experiences, the agent initializes task planning based on the capabilities of the original GPT-4o. We have updated this part at Lines 232-242 in the revised PDF.
>
> ---
>
> **Q4: The sensitivity analysis of the self-verification threshold σ=0.9. How are state embeddings from WM and goal embeddings from LLM aligned for meaningful cosine similarity?**
>
> **Answer:** As shown in the Table below, we performed a sensitivity analysis by running our agent in Minecraft with $10^7$ environment steps for the task "Iron". Experimental results show that the task success rate remains stable when $\sigma \in \[0.875, 0.925\]$, with sharp declines outside this range due to over/under-termination. When $\sigma \textless 0.875$, sub-tasks may not be completed but are misjudged, causing subsequent tasks to fail. When $\sigma \textgreater 0.925$, due to strict self-verification, sub-tasks may be completed but still require re-planning, reducing the task completion rate. We have updated this part at Lines 408-419 in the revised PDF.
>
> |     $\sigma$      | 0.8  | 0.825 | 0.85  | 0.875 |  0.9  | 0.925 | 0.95  | 0.975 |
> | :---------------: | :--: | :---: | :---: | :---: | :---: | :---: | :---: | :---: |
> | Success Rate (SR) | 5.32 | 17.03 | 34.48 | 48.69 | 52.43 | 47.72 | 26.74 | 3.64  |
>
> Similar to MINEDOJO (Fan et al., 2022), we trained a contrastive video-language model pre-trained on the multimodal experience pool. It computes the cosine similarity $\cos(\cdot)$ between an open-vocabulary language goal embedding $\text{Emb}\_{g\_i}$ and an 8-frame video snippet embedding $\text{Emb}\_{s\_t}$, which is used to measure goal attainment. We have updated this part at Lines 271-275 in the revised PDF.
>
> ---
>
> We sincerely thank you for your valuable and constructive feedback. We have integrated these insightful comments into our revised PDF. Unfortunately, the current rating puts us in a difficult position. We sincerely hope our responses are sufficient to address your concerns. We would be very grateful for your timely re-evaluation, as we would strongly prefer to continue with this submission.

---

### Author Response · Authors · 2025-12-03
**Response to ACs**

Dear ACs and PCs,

 **We have successfully addressed all weaknesses and questions raised by the reviewers**. Based on reviewers#d7no Q4, reviewers#ZkBX W2 and W3, reviewers#Khpi W2 and W5, we need to supplement a large number of ablation experiments and sensitivity analysis. Each experiment requires running $10^7$ steps, approximately 2.7 days on a single A100 GPU (the best baseline DreamerV3 requires 7 days). **We use all the lab's computing resources to run a total of 28 rounds of experiments to complete them all before the deadline**. However, due to an OpenReview bug, the reviewers can not respond to my rebuttal. **I sincerely hope that the ACs can carefully evaluate whether our rebuttal has addressed each of the reviewer's concerns point by point.** We express our gratitude once again to all reviewers, ACs and PCs for their thoughtful and valuable feedback.

---

We are grateful that reviewers recognized EvoAgent’s significant contributions:

- **Motivation and Significance** (All Reviewers): Reviewer#d7no highlighted that the paper “**important problem formulation**; this problem is critical for real-world deployment”. Reviewer#ZkBX emphasized that the paper “ addresses the limitations of existing methods; **a significant step forward** for long-horizon task solving.”  Reviewer#Khpi emphasized that the paper "EvoAgent **addresses the challenges** of experience dependency and catastrophic forgetting in long-horizon tasks".
- **Novel Methodological Design** (All Reviewers): Reviewer#d7no appreciated that “**novel continual world model design**; maximally updates environmental understanding efficiently.” Reviewer#ZkBX recognized the originality in introducing “ **a novel framework** designed to address the challenges of long-horizon (LH) task completion in open-ended environments“ and praised "**a compelling and timely contribution**.” Review#Khpi  appreciated that "**the novel self-evolving agent framework**; **strong technical integration**; the proposed planner–controller–reflector loop forms a coherent self-improving mechanism, addressing the limitation of one-directional learning in prior agents."
- **Comprehensive and Convincing Experiments** (All Reviewers): Reviewer#d7no appreciated that “**sound experimental design and strong empirical results**”. Reviewer#ZkBX gave it extremely high praise: "**the experimental evaluation is thorough and convincing**; the use of the challenging Minecraft benchmark provides **strong evidence** for the method's effectiveness and generalization capability". Reviewer#Khpi also highly praised "**comprehensive and convincing experiments**."
- **Clarity and Presentation** (Reviewers#Khpi): Reviewer#Khpi rated the presentation as “excellent,” stating the paper is “**Clear structure and logical presentation**: The paper is well-organized, **with a coherent narrative from motivation to validation**, making its technical and empirical contributions easy to follow.”

---

All reviewers praise the paper's novelty and convincing experiments, primarily asking questions about detailed descriptions, ablation experiments, and computational costs. We have **successfully addressed all weaknesses and questions raised by the reviewers**. We hope it can be accepted by ICLR 2026.



Best Regards,

All Authors

---

> ### Author Response · Authors · 2025-12-04
> **Summary of Rebuttal**
>
> Dear ACs and PCs,
>
>  **We have successfully addressed all weaknesses and questions raised by the reviewers**.
>
> ---
>
> ### Additional Experiments
>
> - **Self-verification threshold analysis** (Reviewer#d7no, #ZkBX): Conducted analysis with $\sigma=0.8, 0.825, 0.85, 0.875, 0.9, 0.925, 0.95, 0.975$, confirming $\sigma=0.9$ as optimal (52.43% SR) versus $\sigma=0.8$ (5.32% SR)  and $\sigma=0.975$ (3.64% SR), where SR is Success Rate.
> - **Two-stage CL ablation** (Reviewer#ZkBX): Evaluated in Minecraft across five material tasks (Wood to Diamond), confirming EvoAgent with both stages as optimal (10.09%±3.54 SR, Diamond) versus EvoAgent only with stage 1 (5.14%±2.51 SR, Diamond) and EvoAgent only with stage 2 (8.93%±3.73 SR, Diamond), validating that joint task-and-experience selecting reduces invalid data impact and computational overhead.
> - **Subtask selection metrics ablation** (Reviewer#ZkBX): Evaluated four metrics (Relevance, Efficiency, Importance, Completion Rate) for the "Iron" task in Minecraft, improving progressively from 40.16% SR (Relevance only) to 49.37% SR (all four metrics), confirming each metric's role in enhancing final performance.
> - **Experience selection metrics ablation** (Reviewer#ZkBX): Evaluated three metrics (TD-Error, Gradient Norm, and Information Gain) for the "Iron" task in Minecraft, improving progressively from 42.72% SR (TD-Error only) to 50.43% SR (all three metrics), demonstrating each metric contributes to more effective experience selection.
> - **World model updating components ablation** (Reviewer#ZkBX): Evaluated the impact of curriculum loss and regularization components for the "Iron" task in Minecraft, improving from 48.61% SR (curriculum loss only) to 50.92% SR (both), confirming regularization's role in mitigating overfitting and enhancing learning stability.
> - **Multimodal experience improving the planner analysis** (Reviewer#ZkBX): Conducted in Minecraft for the “Iron” task, comparing EvoAgent with an experience-free planner (2.58% SR), EvoAgent with a text-only experience planner (6.34% SR), and EvoAgent with a multimodal experience planner (51.07% SR), validating that multimodal experiential feedback is crucial for maximizing planner capability.
> - **Ineffective actions quantitative analysis** (Reviewer#Khpi): Compared EvoAgent with existing outstanding algorithms Jarvis-1, DreamerV3, LS-Imagine, and Optimus-1, improving the average exploration efficiency by 603.28% in the Diamond task, demonstrating that EvoAgent can greatly reduce ineffective exploration.
> - **New comparative experiment** (Reviewer#Khpi): Following reviewer suggestions, compared EvoAgent with LS-Imagine in Minecraft across five material tasks (Wood to Diamond), achieving a success rate (4.36% LS-Imagine vs. 17.36% EvoAgent) and exploration efficiency (4.19% LS-Imagine vs. 26.83% EvoAgent) in the Diamond task, validating that EvoAgent achieves optimal performance.
>
> ---
>
> ### Paper Description
>
> - **Based on the reviewers' comments, we have comprehensively updated the description in the revised PDF**, such as: when to and how to fine-tune the planner (Reviewer#d7no Q3, Reviewer#ZkBX W3-Q4); the cost analysis of LLM API calls (Reviewer#d7no W3, Reviewer#Khpi W6); Figure 2 requiring a clear temporal pipeline (Reviewer#d7no W2.2); the computational overhead (Reviewer#ZkBX Q1); the reward predictor details (Reviewer#Khpi W3), etc.
>
> ---
>
> The reviewers' comments made the contributions of this paper clearer, and **we express our gratitude once again to all reviewers, ACs, and PCs for their thoughtful and valuable feedback**. I sincerely hope this paper can be accepted by ICLR 2026.

---

### Meta-Review · Area_Chair_gkQG · 2026-01-03

**Summary:**

This paper introduces EvoAgent, a self-evolving agent framework that integrates a continual world model (WM) with a planner-controller-reflector loop to address challenges in long-horizon (LH) task completion in open-ended environments. The key idea is to allow the agent to self-plan, self-control, and self-reflect to autonomously update its world knowledge and reduce dependence on curated human experience or curricula.

While reviewers acknowledged the importance of the addressed problem, and found the experimental results on Minecraft and Atari impressive, the overall evaluation was mixed, and several major concerns were raised:

**Clarity and Writing Issues**: Multiple reviewers (ZkBX, d7no) pointed out the paper’s lack of clarity, including ambiguous notation, confusing visualizations (e.g., Figure 2), and poor citation formatting.

**Unclear Novelty and Contribution Attribution**: Reviewer ZkBX questioned which components are novel, how they interact, and whether the contributions were incremental combinations of existing ideas (e.g., LoRA fine-tuning, curriculum learning, RSSM, self-verification).

**Experimental Overhead and Fairness of Comparison**: Several reviewers (ZkBX, Khpi) requested clarification on the computational cost (especially LLM API usage), and whether the added complexity (e.g., dual-stage curriculum, multimodal experience, GPT-based planner) was justified by the performance gains.

**Insufficient Methodological Detail**: Concerns were raised regarding the reward predictor architecture, visual encoding, and fine-tuning strategy (e.g., using only failed subtasks). These raised doubts about the soundness and reproducibility of the method (Khpi, d7no).

**Incomplete Baseline Comparisons**: Reviewer Khpi noted the absence of recent strong baselines such as LS-Imagine, which was later included in the rebuttal. However, the effectiveness of EvoAgent relative to these newer methods remains debatable.

While the paper tackles an important and challenging problem in long-horizon embodied learning and includes strong empirical results and ablations, the methodological contribution is unclear, and the paper lacks focus and clarity. The approach is heavily engineered, and the improvements—while measurable—may not justify the added complexity. The authors made a strong rebuttal effort, but fundamental concerns about novelty, clarity, and generality remain.

**Final Recommendation: Reject**\
**Justification**: Despite promising results and extensive experiments, the submission does not clearly establish a novel core insight, and the method’s contribution is primarily system-level. The paper is not yet ready for publication at ICLR in its current form.

**Reviewer Concerns:**

**Addressed Concerns**\
The authors responded with significant additional experiments and clarifications, which are commendable. The following concerns were reasonably addressed:

***Ablation Studies and Sensitivity Analysis***: The authors provided comprehensive ablations for curriculum stages, selection metrics (TD-error, gradient norm, etc.), self-verification threshold, and planner architecture (text-only vs. multimodal). These strengthened the empirical support for the design choices.

***LLM Cost and Efficiency***: The rebuttal included API usage statistics and inference time comparisons (e.g., EvoAgent runs faster than DreamerV3), addressing concerns about computational cost.

***Experimental Comparison with LS-Imagine***: The authors added new experiments comparing EvoAgent with LS-Imagine, showing improved performance and exploration efficiency, especially on harder tasks.

***Clarification of Visual Encoding and Reward Predictor***: The authors described the tokenization and projection pipeline for multimodal data and provided more details on the goal-conditioned reward predictor.

**Remaining Concerns**\
Despite the thorough rebuttal, key concerns remain unresolved and impact the overall evaluation:

***Clarity and Focus of the Paper***: Even after revisions, the paper remains dense and overloaded with components. The core novelty is still difficult to isolate. The planner-controller-reflector loop is not clearly modularized in terms of what is new vs. reused from prior work.

***Over-reliance on Engineering Design***: The method appears to be a heavy integration of existing techniques (LLMs + RSSM + LoRA fine-tuning + curriculum learning), rather than a fundamentally new algorithmic contribution. The contributions are more system-level than theoretical or methodological.

***Marginal Gains vs. Complexity***: While EvoAgent shows improvement, it introduces significant additional complexity (e.g., dual-stage reflection, multimodal experience tracking, planner fine-tuning on failures), which raises concerns about scalability and practicality.

***Generalizability and Transferability***: Although the authors claim potential extension to robotics, no experiments or results outside of Minecraft and Atari are provided. The method’s real-world applicability remains speculative.

**Reviewer Scores:**

**Reviewer d7no (Initial Score: 6)**: Raised concerns about fine-tuning logic, subtask failure handling, and cost analysis. The rebuttal addressed most technical questions, but the core concern about method complexity and soundness likely remains. Final Score Prediction: 6 (unchanged)

**Reviewer ZkBX (Initial Score: 4)**: Highlighted issues with writing, unclear novelty, and ablation coverage. The rebuttal addressed these with added experiments and clarifications, but overall complexity and limited conceptual novelty appear unresolved. Final Score Prediction: 4 (unchanged)

**Reviewer Khpi (Initial Score: 4)**: Acknowledged the method’s strengths but raised concerns about reward predictor, visual encoding, and incomplete comparisons. While the rebuttal was thorough, the method still may not meet the bar for acceptance. Final Score Prediction: 4 (unchanged)

---

### Decision · Program_Chairs · 2026-01-26

Reject